# EFFICIENT EPISODIC MEMORY UTILIZATION OF COOPERATIVE MULTI-AGENT REINFORCEMENT LEARNING

**Hyungho Na[1], Yunkyeong Seo[1] & Il-Chul Moon[1,2]**
[1]Korea Advanced Institute of Science and Technology (KAIST), [2]summary.ai
{gudgh723}@gmail.com,{tjdbsrud,icmoon}@kaist.ac.kr

## ABSTRACT

In cooperative multi-agent reinforcement learning (MARL), agents aim to achieve a common goal, such as defeating enemies or scoring a goal. Existing MARL algorithms are effective but still require significant learning time and often get trapped in local optima by complex tasks, subsequently failing to discover a goal-reaching policy. To address this, we introduce Efficient episodic Memory Utilization (EMU) for MARL, with two primary objectives: (a) accelerating reinforcement learning by leveraging semantically coherent memory from an episodic buffer and (b) selectively promoting desirable transitions to prevent local convergence. To achieve (a), EMU incorporates a trainable encoder/decoder structure alongside MARL, creating coherent memory embeddings that facilitate exploratory memory recall. To achieve (b), EMU introduces a novel reward structure called episodic incentive based on the desirability of states. This reward improves the TD target in Q-learning and acts as an additional incentive for desirable transitions. We provide theoretical support for the proposed incentive and demonstrate the effectiveness of EMU compared to conventional episodic control. The proposed method is evaluated in StarCraft II and Google Research Football, and empirical results indicate further performance improvement over state-of-the-art methods. Our code is available at: https://github.com/HyunghoNa/EMU.

## 1 INTRODUCTION

Recently, cooperative MARL has been adopted to many applications, including traffic control (Wiering et al., 2000), resource allocation (Dandanov et al., 2017), robot path planning (Wang et al., 2020a), and production systems  (Dittrich & Fohlmeister, 2020), etc. In spite of these successful applications, cooperative MARL still faces challenges in learning proper coordination among multiple agents because of the partial observability and the interaction between agents during training.

To address these challenges, the framework of centralized training and decentralized execution (CTDE) (Oliehoek et al., 2008; Oliehoek & Amato, 2016; Gupta et al., 2017) has been proposed. CTDE enables a decentralized execution while fully utilizing global information during centralized training, so CTDE improves policy learning by accessing to global states at the training phase. Especially, value factorization approaches (Sunehag et al., 2017; Rashid et al., 2018; Son et al., 2019; Yang et al., 2020; Rashid et al., 2020; Wang et al., 2020b) maintain the consistency between individual and joint action selection, achieving the state-of-the-art performance on difficult multi-agent tasks, such as StarCraft II Multi-agent Challenge (SMAC) (Samvelyan et al., 2019). However, learning optimal policy in MARL still requires a long convergence time due to the interaction between agents, and the trained models often fall into local optima, particularly when agents perform complex tasks (Mahajan et al., 2019). Hence, researchers present a committed exploration mechanism under this CTDE training practice (Mahajan et al., 2019; Yang et al., 2019; Wang et al., 2019; Liu et al., 2021) with the expectation to find episodes escaping from the local optima.

Despite the required exploration in MARL with CTDE, recent works on episodic control emphasize the exploitation of episodic memory to expedite reinforcement learning. Episodic control (Lengyel & Dayan, 2007; Blundell et al., 2016; Lin et al., 2018; Pritzel et al., 2017) memorizes explored states and their best returns from experience in the episodic memory, to converge on the best policy. Recently, this episodic control has been adopted to MARL (Zheng et al., 2021), and this episodic

control case shows faster convergence than the learning without such memory. Whereas there are merits from episodic memory and control from its utilization, there exists a problem of determining which memories to recall and how to use them, to efficiently explore from the memory. According to Blundell et al. (2016); Lin et al. (2018); Zheng et al. (2021), the previous episodic control generally utilizes a random projection to embed global states, but this random projection hardly makes the semantically similar states close to one another in the embedding space. In this case, exploration will be limited to a narrow distance threshold. However, this small threshold leads to inefficient memory utilization because the recall of episodic memory under such small thresholds retrieves only the same state without consideration of semantic similarity from the perspective of goal achievement. Additionally, the naive utilization of episodic control on complex tasks involves the risk of converging to local optima by repeatedly revisiting previously explored states, favoring exploitation over exploration.

**Contribution.** This paper presents an **E**fficient episodic **M**emory **U**tilization for multi-agent reinforcement learning (EMU), a framework to selectively encourage desirable transitions with semantic memory embeddings.

- **Efficient memory embedding:** When generating features of a global state for episodic memory (Figure 1(b)), we adopt an encoder/decoder structure where 1) an encoder embeds a global state conditioned on timestep into a low-dimensional feature and 2) a decoder takes this feature as an input conditioned on the timestep to predict the return of the global state. In addition, to ensure smoother embedding space, we also consider the reconstruction of the global state when training the decoder to predict its return. To this end, we develop **d**eterministic **C**onditional **A**uto**E**ncoder (dCAE) (Figure 1(c)). With this structure, important features for overall return can be captured in the embedding space. The proposed embedding contains semantic meaning and thus guarantees a gradual change of feature space, making the further exploration on memory space near the given state, i.e., efficient memory utilization.

- **Episodic incentive generation:** While the semantic embedding provides a space to explore, we still need to identify promising state transitions to explore. Therefore, we define a ***desirable trajectory*** representing the highest return path, such as destroying all enemies in SMAC or scoring a goal in Google Research Football (GRF) (Kurach et al., 2020). States on this trajectory are marked as desirable in episodic memory, so we could incentivize the exploration on such states according to their desirability. We name this incentive structure as an episodic incentive (Figure 1(d)), encouraging desirable transitions and preventing convergence to unsatisfactory local optima. We provide theoretical analyses demonstrating that this episodic incentive yields a better gradient signal compared to conventional episodic control.

We evaluate EMU on SMAC and GRF, and empirical results demonstrate that the proposed method achieves further performance improvement compared to the state-of-art baseline methods. Ablation studies and qualitative analyses validate the propositions made by this paper.

## 2 PRELIMINARY

### 2.1 DECENTRALIZED POMDP

A fully cooperative multi-agent task can be formalized by following the Decentralized Partially Observable Markov Decision Process (Dec-POMDP) (Oliehoek & Amato, 2016), $G = \langle I, S, A, P, R, \Omega, O, n, \gamma \rangle$, where $I$ is the finite set of $n$ agents; $s \in S$ is the true state of the environment; $a_i \in A$ is the $i$-th agent's action forming the joint action $\boldsymbol{a} \in A^n$; $P(s'|s, \boldsymbol{a})$ is the state transition function; $R$ is a reward function $r = R(s, \boldsymbol{a}, s') \in \mathbb{R}$; $\Omega$ is the observation space; $O$ is the observation function generating an observation for each agent $o_i \in \Omega$; and finally, $\gamma \in [0, 1)$ is a discount factor. At each timestep, an agent has its own local observation $o_i$, and the agent selects an action $a_i \in A$. The current state $s$ and the joint action of all agents $\boldsymbol{a}$ lead to a next state $s'$ according to $P(s'|s, \boldsymbol{a})$. The joint variable of $s$, $\boldsymbol{a}$, and $s'$ will determine the identical reward $r$ across the multi-agent group. In addition, similar to Hausknecht & Stone (2015); Rashid et al. (2018), each agent utilizes a local action-observation history $\tau_i \in T \equiv (\Omega \times A)$ for its policy $\pi_i(a|\tau_i)$, where $\pi : T \times A \rightarrow [0, 1]$.

## 2.2 DESIRABILITY AND DESIRABLE TRAJECTORY

**Definition 1.** (Desirability and Desirable Trajectory) For a given threshold return $R_{thr}$ and a trajectory $\mathcal{T} := \{s_0, \boldsymbol{a_0}, r_0, s_1, \boldsymbol{a_1}, r_1, ..., s_T\}$, $\mathcal{T}$ is considered as a desirable trajectory, denoted as $\mathcal{T}_\xi$, when an episodic return is $R_{t=0} = \Sigma_{t'=t}^{T-1} r_{t'} \geq R_{\text{thr}}$. A binary indicator $\xi(\cdot)$ denotes the desirability of state $s_t$ as $\xi(s_t) = 1$ when $s_t \in \forall \mathcal{T}_\xi$.

In cooperative MARL tasks, such as SMAC and GRF, the total amount of rewards from the environment within an episode is often limited as $R_{\max}$, which is only given when cooperative agents achieve a common goal. In such a case, we can set $R_{thr} = R_{\max}$. For further description of cooperative MARL, please see Appendix A.

## 2.3 EPISODIC CONTROL IN MARL

Episodic control was introduced from the analogy of a brain's hippocampus for memory utilization (Lengyel & Dayan, 2007). After the introduction of deep Q-network, Blundell et al. (2016) adopted this idea of episodic control to the model-free setting by storing the highest return of a given state, to efficiently estimate the Q-values of the state. This recalling of the high-reward experiences helps to increase sample efficiency and thus expedites the overall learning process (Blundell et al., 2016; Pritzel et al., 2017; Lin et al., 2018). Please see Appendix A for related works and further discussions.

At timestep $t$, let us define a global state as $s_t$. When utilizing episodic control, instead of directly using $s_t$, researchers adopt a state embedding function $f_\phi(s) : S \to \mathbb{R}^k$ to project states toward a $k$-dimensional vector space. With this projection, a representation of global state $s_t$ becomes $x_t = f_\phi(s_t)$. The episodic control memorizes $H(f_\phi(s_t))$, i.e., the highest return of a given global state $s_t$, in episodic buffer $\mathcal{D}_E$ (Pritzel et al., 2017; Lin et al., 2018; Zheng et al., 2021). Here, $x_t$ is used as a key to the highest return, $H(x_t)$; as a key-value pair in $\mathcal{D}_E$. The episodic control in Lin et al. (2018) updates $H(x_t)$ with the following rules.

$$H(x_t) = \begin{cases} \max\{H(\hat{x}_t), R_t(s_t, \boldsymbol{a_t})\}, & \text{if } ||\hat{x}_t - x_t||_2 < \delta \\ R_t(s_t, \boldsymbol{a_t}), & \text{otherwise}, \end{cases} \tag{1}$$

where $R_t(s_t, \boldsymbol{a_t})$ is the return of a given $(s_t, \boldsymbol{a_t})$; $\delta$ is a threshold value of state-embedding difference; and $\hat{x}_t = f_\phi(\hat{s}_t)$ is $x_t = f_\phi(s_t)$'s nearest neighbor in $\mathcal{D}_E$. If there is no similar projected state $\hat{x}_t$ such that $||\hat{x}_t - x_t||_2 < \delta$ in the memory, then $H(x_t)$ keeps the current $R_t(s_t, \boldsymbol{a_t})$. Leveraging the episodic memory, EMC (Zheng et al., 2021) presents the one-step TD memory target $Q_{EC}(f_\phi(s_t), \boldsymbol{a_t})$ as

$$Q_{EC}(f_\phi(s_t), \boldsymbol{a_t}) = r_t(s_t, \boldsymbol{a_t}) + \gamma H(f_\phi(s_{t+1})). \tag{2}$$

Then, the loss function $L_\theta^{EC}$ for training can be expressed as the weighted sum of one-step TD error and one-step TD memory error, i.e., Monte Carlo (MC) inference error, based on $Q_{EC}(f_\phi(s_t), \boldsymbol{a_t})$.

$$L_\theta^{EC} = (y(s, \boldsymbol{a}) - Q_{tot}(s, \boldsymbol{a}; \theta))^2 + \lambda(Q_{EC}(f_\phi(s), \boldsymbol{a}) - Q_{tot}(s, \boldsymbol{a}; \theta))^2, \tag{3}$$

where $y(s, \boldsymbol{a})$ is one-step TD target; $Q_{tot}$ is the joint Q-value function parameterized by $\theta$; and $\lambda$ is a scale factor.

**Problem of the conventional episodic control with random projection**   Random projection is useful for dimensionality reduction as it preserves distance relationships, as demonstrated by the Johnson-Lindenstrauss lemma (Dasgupta & Gupta, 2003). However, a random projection adopted for $f_\phi(s)$ hardly has a semantic meaning in its embedding $x_t$, as it puts random weights on the state features without considering the patterns of determining the state returns. Additionally, when recalling the memory from $\mathcal{D}_E$, the projected state $x_t$ can abruptly change even with a small change of $s_t$ because the embedding is not being regulated by the return. This results in a sparse selection of semantically similar memories, i.e. similar states with similar or better rewards. As a result, conventional episodic control using random projection only recalls identical states and relies on its own Monte-Carlo (MC) return to regulate the one-step TD target inference, limiting exploration of nearby states on the embedding space.

The problem intensifies when the high-return states in the early training phase are indeed local optima. In such cases, the naive utilization of episodic control is prone to converge on local minima. As a result, for the super hard tasks of SMAC, EMC (Zheng et al., 2021) had to decrease the magnitude of this regularization to almost zero, i.e., not considering episodic memories anymore.

## 3 METHODOLOGY

This section introduces **E**fficient episodic **M**emory **U**tilization (EMU) (Figure 1). We begin by explaining how to construct **(1) semantic memory embeddings** to better utilize the episodic memory, which enables memory recall of similar, more promising states. To further improve memory utilization, as an alternative to the conventional episodic control, we propose **(2) episodic incentive** that selectively encourages desirable transitions while preventing local convergence towards undesirable trajectories.

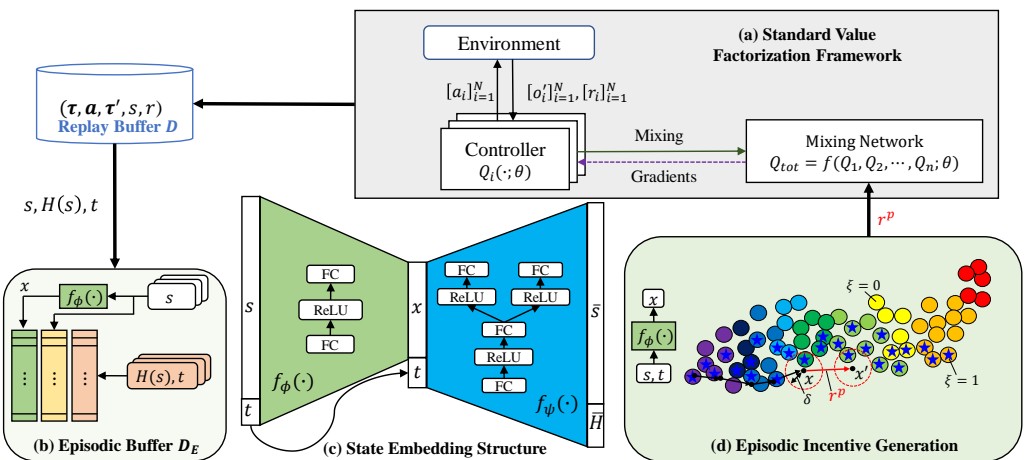

Figure 1: Overview of EMU framework.

### 3.1 SEMANTIC MEMORY EMBEDDING

**Episodic Memory Construction** To address the problems of a random projection adopted in episodic control, we propose a trainable embedding function $f_\phi(s)$ to learn the state embedding patterns affected by the highest return. The problem of a learnable embedding network $f_\phi$ is that the match between $H(f_\phi(s_t))$ and $s_t$ breaks whenever $f_\phi$ is updated. Hence, we save the global state $s_t$ as well as a pair of $H_t$ and $x_t$ in $\mathcal{D}_E$, so that we can update $x = f_\phi(s)$ whenever $f_\phi$ is updated. In addition, we store the desirability $\xi$ of $s_t$ according to Definition 1. Appendix E.1 illustrates the details of memory construction proposed by this paper.

**Learning framework for State Embedding** When training $f_\phi(s_t)$, it is critical to extract important features of a global state that affect its value, i.e., the highest return. Thus, we additionally adopt a decoder structure $\bar{H}_t = f_\psi(x_t)$ to predict the highest return $H_t$ of $s_t$. We call this embedding function as **EmbNet**, and its learning objective of $f_\phi$ and $f_\psi$ can be written as

$$\mathcal{L}(\phi, \psi) = (H_t - f_\psi(f_\phi(s_t)))^2. \tag{4}$$

When constructing the embedding space, we found that an additional consideration of reconstruction of state $s$ conditioned on timestep $t$ improves the quality of feature extraction and constitutes a smoother embedding space. To this end, we develop the deterministic conditional autoencoder (**dCAE**), and the corresponding loss function can be expressed as

$$\mathcal{L}(\phi, \psi) = \left(H_t - f_\psi^H(f_\phi(s_t|t)|t)\right)^2 + \lambda_{rcon}||s_t - f_\psi^s(f_\phi(s_t|t)|t)||_2^2, \tag{5}$$

where $f_\psi^H$ predicts the highest return; $f_\psi^s$ reconstructs $s_t$; $\lambda_{rcon}$ is a scale factor. Here, $f_\psi^H$ and $f_\psi^s$ share the lower part of networks as illustrated in Figure 1(c). Appendix C.1 presents the details of network structure of $f_\phi$ and $f_\psi$, and Algorithm 1 in Appendix C.1 presents the learning framework for $f_\phi$ and $f_\psi$. This training is conducted periodically in parallel to the RL policy learning on $Q_{tot}(\cdot; \theta)$.

Figure 2 illustrates the result of t-SNE (Van der Maaten & Hinton, 2008) of 50K samples of $x \in \mathcal{D}_E$ out of 1M memory data in training for `3s_vs_5z` task of SMAC. Unlike supervised learning with label data, there is no label for each $x_t$. Thus, we mark $x_t$ with its pair of the highest return $H_t$. Compared to a random projection in Figure 2(a), $x_t$ via $f_\phi$ is well-clustered, according to the similarity of the embedded state and its return. This clustering of $x_t$ enables us to safely select

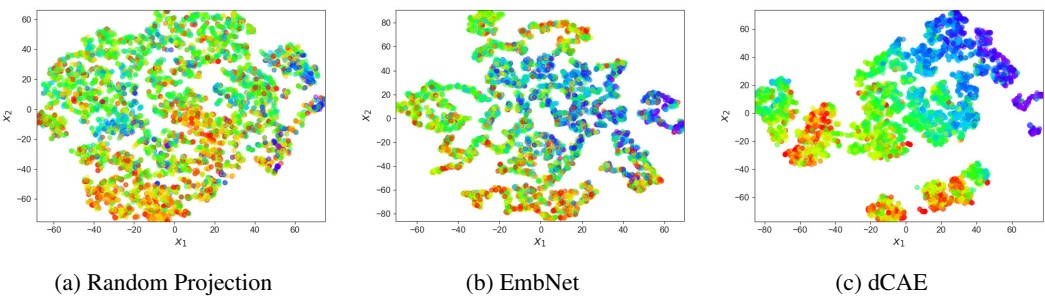

(a) Random Projection      (b) EmbNet      (c) dCAE

Figure 2: t-SNE of sampled embedding $x \in \mathcal{D}_E$. Colors from red to purple (rainbow) represent from low return to high return.

episodic memories around the key state $s_t$, which constitutes efficient memory utilization. This memory utilization expedites learning speed as well as encourages exploration to a more promising state $\hat{s}_t$ near $s_t$. Appendix F illustrates how to determine $\delta$ of Eq. 1 in a memory-efficient way.

## 3.2 EPISODIC INCENTIVE

With the learnable memory embedding for an efficient memory recall, how to use the selected memories still remains a challenge because a naive utilization of episodic memory is prone to converge on local minima. To solve this issue, we propose a new reward structure called **episodic incentive** $r^p$ by leveraging the desirability $\xi$ of states in $\mathcal{D}_E$. Before deriving the episodic incentive $r^p$, we first need to understand the characteristics of episodic control. In this section, we denote the joint Q-function $Q_{tot}(\cdot; \theta)$ simply as $Q_\theta$ for conciseness.

**Theorem 1.** *Given a transition $(s, \boldsymbol{a}, r, s')$ and $H(x')$, let $L_\theta$ be the Q-learning loss with additional transition reward, i.e., $L_\theta := (y(s, \boldsymbol{a}) + r^{EC}(s, \boldsymbol{a}, s') - Q_{tot}(s, \boldsymbol{a}; \theta))^2$ where $r^{EC}(s, \boldsymbol{a}, s') := \lambda(r(s, \boldsymbol{a}) + \gamma H(x') - Q_\theta(s, \boldsymbol{a}))$, then $\nabla_\theta L_\theta = \nabla_\theta L_\theta^{EC}$. (Proof in Appendix B.1)*

As Theorem 1 suggests, we can generate the same gradient signal as the episodic control by leveraging the additional transition reward $r^{EC}(s, \boldsymbol{a}, s')$. However, $r^{EC}(s, \boldsymbol{a}, s')$ accompanies a risk of local convergence as discussed in Section 2.3. Therefore, instead of applying $r^{EC}(s, \boldsymbol{a}, s')$, we propose the episodic incentive $r^p := \gamma \hat{\eta}(s')$ that provides an additional reward for the desirable transition $(s, \boldsymbol{a}, r, s')$, such that $\xi(s') = 1$. Here, $\hat{\eta}(s')$ estimates $\eta^*(s')$, which represents the difference between the true value $V^*(s')$ of $s'$ and the predicted value via target network $\max_{\boldsymbol{a}'} Q_{\theta^-}(s', \boldsymbol{a}')$, defined as

$$\eta^*(s') := V^*(s') - \max_{\boldsymbol{a}'} Q_{\theta^-}(s', \boldsymbol{a}'). \tag{6}$$

Note that we do not know $V^*(s')$ and subsequently $\eta^*(s')$. To accurately estimate $\eta^*(s')$ with $\hat{\eta}(s')$, we use the expected value considering the current policy $\pi_\theta$ as $\hat{\eta}(s') := \mathbb{E}_{\pi_\theta}[\eta(s')]$ where $\eta \in [0, \eta_{\max}(s')]$ for $s' \sim P(s'|s, \boldsymbol{a} \sim \pi_\theta)$. Here, $\eta_{\max}(s')$ can be reasonably approximated by using $H(f_\phi(s'))$ in $\mathcal{D}_E$. Then, with the count-based estimation $\hat{\eta}(s')$, episodic incentive $r^p$ can be expressed as

$$r^p = \gamma \hat{\eta}(s') = \gamma \mathbb{E}_{\pi_\theta}[\eta(s')] \simeq \gamma \frac{N_\xi(s')}{N_{call}(s')} \eta_{\max}(s') = \gamma \frac{N_\xi(s')}{N_{call}(s')}(H(f_\phi(s')) - \max_{a'} Q_{\theta^-}(s', a')), \tag{7}$$

where $N_{call}(s')$ is the number of visits on $\hat{x}' = \text{NN}(f_\phi(s')) \in \mathcal{D}_E$; and $N_\xi$ is the number of desirable transition from $\hat{x}'$. Here, $\text{NN}(\cdot)$ represents a function for selecting the nearest neighbor. From Theorem 1, the loss function adopting episodic control with an alternative transition reward $r^p$ instead of $r^{EC}$ can be expressed as

$$L_\theta^p = (r(s, \boldsymbol{a}) + r^p + \gamma \max_{\boldsymbol{a}'} Q_{\theta^-}(s', \boldsymbol{a}') - Q_\theta(s, \boldsymbol{a}))^2. \tag{8}$$

Then, the gradient signal of the one-step TD inference loss $\nabla_\theta L_\theta^p$ with the episodic reward $r^p = \gamma \hat{\eta}(s')$ can be written as

$$\nabla_\theta L_\theta^p = -2 \nabla_\theta Q_\theta(s, a)(\Delta \varepsilon_{TD} + r^p) = -2 \nabla_\theta Q_\theta(s, a)(\Delta \varepsilon_{TD} + \gamma \frac{N_\xi(s')}{N_{call}(s')} \eta_{\max}(s')), \tag{9}$$

where $\Delta\varepsilon_{TD} = r(s,a) + \gamma\max_{a'}Q_{\theta^-}(s',a') - Q_\theta(s,a)$ is one-step inference TD error. Here, the gradient signal $\nabla_\theta L_\theta^p$ with the proposed episodic reward $r^p$ can accurately estimate the optimal gradient signal as follows.

**Theorem 2.** *Let $\nabla_\theta L_\theta^* = -2\nabla_\theta Q_\theta(s,a)(\Delta\varepsilon_{TD}^*)$ be the optimal gradient signal with the true one step TD error $\Delta\varepsilon_{TD}^* = r(s,a) + \gamma V^*(s') - Q_\theta(s,a)$. Then, the gradient signal $\nabla_\theta L_\theta^p$ with the episodic incentive $r^p$ converges to the optimal gradient signal as the policy converges to the optimal policy $\pi_\theta^*$, i.e., $\nabla_\theta L_\theta^p \to \nabla_\theta L_\theta^*$ as $\pi_\theta \to \pi_\theta^*$. (Proof in Appendix B.2)*

Theorem 2 also implies that there exists a certain bias in $\nabla_\theta L_\theta^{EC}$ as described in Appendix B.2. Besides the property of convergence to the optimal gradient signal presented in Theorem 2, the episodic incentive has the following additional characteristics. (1) The episodic incentive is only applied to the desirable transition. We can simply see that $r^p = \gamma\hat{\eta} = \gamma\mathbb{E}_{\pi_\theta}[\eta] \simeq \gamma\eta_{max}N_\xi/N_{call}$ and if $\xi(s') = 0$ then $N_\xi = 0$, yielding $r^p \to 0$. Subsequently, (2) there is no need to adjust a scale factor by the task complexity. (3) The episodic incentive can reduce the risk of overestimation by considering the

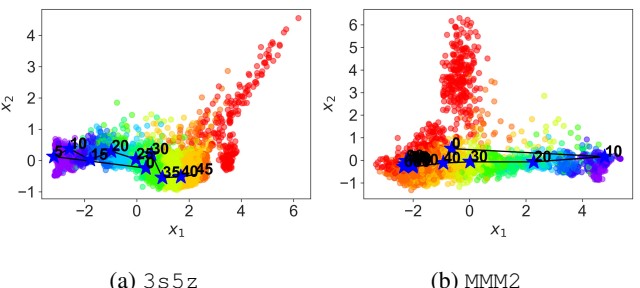

(a) 3s5z          (b) MMM2

Figure 3: Episodic incentive. Test trajectories are plotted on the embedded space with sampled memories in $\mathcal{D}_E$, denoted with dotted markers. Star markers and numbers represent the desirability of state and timestep in the episode, respectively. Color represents the same semantics as Figure 2.

expected value of $\mathbb{E}_{\pi_\theta}[\eta]$. Instead of considering the optimistic $\eta_{max}$, the count-based estimation $r^p = \gamma\hat{\eta} = \gamma\mathbb{E}_{\pi_\theta}[\eta]$ can consider the randomness of the policy $\pi_\theta$. Figure 3 illustrates how the episodic incentive works with the desirability stored in $\mathcal{D}_E$ constructed by Algorithm 2 presented in Appendix E.1. In Figure 3 as we intended, high-value states (at small timesteps) are clustered close to the purple zone, while low-value states (at large timesteps) are located in the red zone.

### 3.3 OVERALL LEARNING OBJECTIVE

To construct the joint Q-function $Q_{tot}$ from individual $Q_i$ of the agent $i$, any form of mixer can be used. In this paper, we mainly adopt the mixer presented in QPLEX (Wang et al., 2020b) similar to Zheng et al. (2021), which guarantees the complete Individual-Global-Max (IGM) condition (Son et al., 2019; Wang et al., 2020b). Considering any intrinsic reward $r^c$ encouraging an exploration (Zheng et al., 2021) or diversity (Chenghao et al., 2021), the final loss function for the action policy learning from Eq. 8 can be extended as

$$\mathcal{L}_\theta^p = \left(r(s,\boldsymbol{a}) + r^p + \beta_c r^c + \gamma\max_{\boldsymbol{a}'}Q_{tot}(s',\boldsymbol{a}';\theta^-) - Q_{tot}(s,\boldsymbol{a};\theta)\right)^2, \tag{10}$$

where $\beta_c$ is a scale factor. Note that the episodic incentive $r^p$ can be used in conjunction with any form of intrinsic reward $r^c$ being properly annealed throughout the training. Again, $\theta$ denotes the parameters of networks related to action policy $Q_i$ and the corresponding mixer network to generate $Q_{tot}$. For the action selection via $Q$, we adopt a GRU to encode a local action-observation history $\tau$ presented in 2.1 similar to Sunehag et al. (2017); Rashid et al. (2018); Wang et al. (2020b); but in Eq. 10, we denote equations with $s$ instead of $\tau$ for the coherence with derivation in the previous section. Appendix E.2 presents the overall training algorithm.

## 4 EXPERIMENTS

In this part, we have formulated our experiments with the intention of addressing the following inquiries denoted as Q1-3.

- Q1. How does EMU compare to the state-of-the-art MARL frameworks?
- Q2. How does the proposed state embedding change the embedding space and improve the performance?
- Q3. How does the episodic incentive improve performance?

We conduct experiments on complex multi-agent tasks such as SMAC (Samvelyan et al., 2019) and GRF (Kurach et al., 2020). The experiments compare EMU against EMC adopting episodic control (Zheng et al., 2021). Also, we include notable baselines, such as value-based MARL methods QMIX (Rashid et al., 2018), QPLEX (Wang et al., 2020b), CDS encouraging individual diversity (Chenghao et al., 2021). Particularly, we emphasize that EMU can be combined with any MARL framework, so we present two versions of EMU implemented on original QPLEX and CDS, denoted as EMU (QPLEX) and EMU (CDS), respectively. Appendix C provides further details of experiment settings and implementations, and Appendix D.12 provides the applicability of EMU to single-agent tasks, including pixel-based high-dimensional tasks.

## 4.1 Q1. COMPARATIVE EVALUATION ON STARCRAFT II (SMAC)

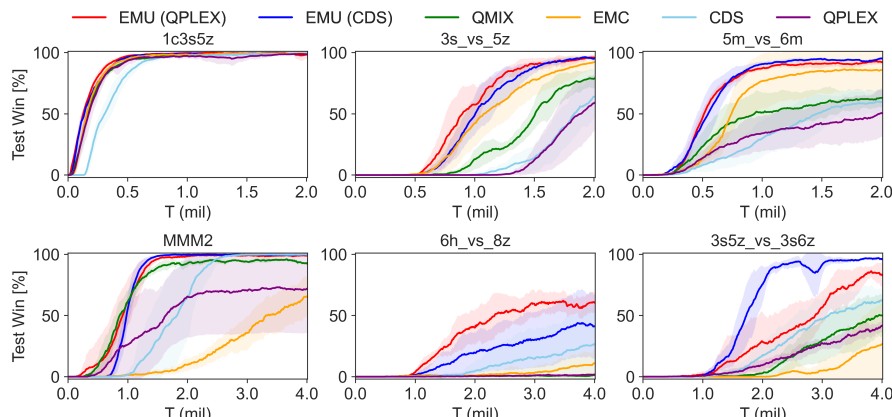

Figure 4: Performance comparison of EMU against baseline algorithms on three **easy and hard** SMAC maps: `1c3s5z`, `3s_vs_5z`, and `5m_vs_6m`, and three **super hard** SMAC maps: `MMM2`, `6h_vs_8z`, and `3s5z_vs_3s6z`.

Figure 4 illustrates the overall performance of EMU on various SMAC maps. The map categorization regarding the level of difficulty follows the practice of Samvelyan et al. (2019). Thanks to the efficient memory utilization and episodic incentive, both EMU (QPLEX) and EMU (CDS) show significant performance improvement compared to their original methodologies. Especially, in **super hard** SMAC maps, the proposed method markedly expedites convergence on optimal policy.

## 4.2 Q1. COMPARATIVE EVALUATION ON GOOGLE RESEARCH FOOTBALL (GRF)

Here, we conduct experiments on GRF to further compare the performance of EMU with other baseline algorithms. In our GRF task, CDS and EMU (CDS) do not utilize the agent's index on observation as they contain the prediction networks while other baselines (QMIX, EMC, QPLEX) use information of the agent's identity in observations. In addition, we do not utilize any additional algorithm, such as prioritized experience replay (Schaul et al., 2015), for all baselines and our method, to expedite learning efficiency. From the experiments, adopting EMU achieves significant performance improvement, and EMU quickly finds the winning or scoring policy at the early learning phase by utilizing semantically similar memory.

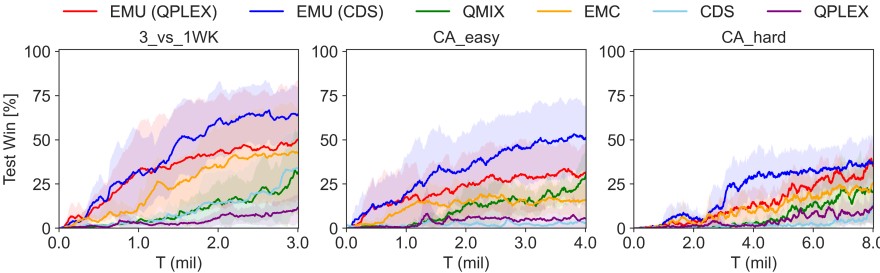

Figure 5: Performance comparison of EMU against baseline algorithms on Google Research Football.

### 4.3 Q2. Parametric and Ablation Study

In this section, we examine how the key hyperparameter $\delta$ and the choice of design for $f_\phi$ affect the performance. To compare the learning quality and performance more quantitatively, we propose a new performance index called *overall win-rate*, $\bar{\mu}_w$. The purpose of $\bar{\mu}_w$ is to consider both training efficiency (speed) and quality (win-rate) for different seed cases (see Appendix D.1 for details). We conduct experiments on selected SMAC maps to measure $\bar{\mu}_w$ according to $\delta$ and design choice for $f_\phi$ such as (1) random projection, (2) **EmbNet** with Eq. 4 and (3) **dCAE** with Eq. 5.

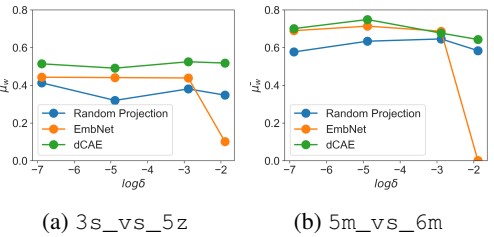

(a) 3s_vs_5z    (b) 5m_vs_6m

Figure 6: $\bar{\mu}_w$ according to $\delta$ and various design choices for $f_\phi$ on SMAC maps.

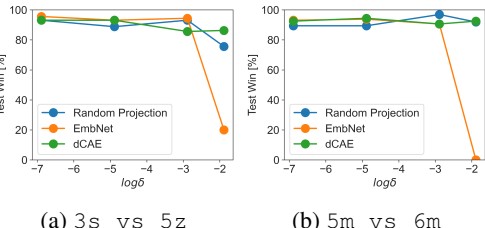

(a) 3s_vs_5z    (b) 5m_vs_6m

Figure 7: Final win-rate according to $\delta$ and various design choices for $f_\phi$ on SMAC maps.

Figure 6 and Figure 7 show $\bar{\mu}_w$ values and test win-rate at the end of training time according to different $\delta$, presented in log-scale. To see the effect of design choice for $f_\phi$ distinctly, we conduct experiments with the conventional episodic control. More data of $\bar{\mu}_w$ is presented in Tables 4 and 5 in Appendix D.2. Figure 6 illustrates that dCAE structure shows the best training efficiency throughout various $\delta$ while achieving the optimal policy as other design choices as presented in Figure 7.

Interestingly, dCAE structure works well with a wider range of $\delta$ than EmbNet. We conjecture that EmbNet can select very different states as exploration if those states have similar return $H$ during training. This excessive memory recall adversely affects learning and fails to find an optimal policy as a result. See Appendix D.2 for detailed analysis and Appendix D.8 for an ablation study on the loss function of dCAE.

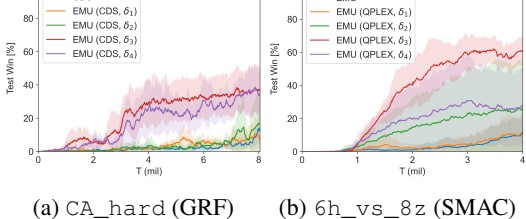

(a) CA_hard (GRF)    (b) 6h_vs_8z (SMAC)

Figure 8: Effect of varying $\delta$ on complex MARL tasks.

Even though a wide range of $\delta$ works well as in Figures 6 and 7, choosing a proper value of $\delta$ in more difficult MARL tasks significantly improves the overall learning performance. Figure 8 shows the learning curve of EMU according to $\delta_1 = 1.3e^{-7}$, $\delta_2 = 1.3e^{-5}$, $\delta_3 = 1.3e^{-3}$, and $\delta_4 = 1.3e^{-2}$. In super hard MARL tasks such as 6h_vs_8z in SMAC and CA_hard in GRF, $\delta_3$ shows the best performance compared to other $\delta$ values. This is consistent with the value suggested in Appendix F, where $\delta$ is determined in a memory-efficient way. Further parametric study on $\delta$ and $\lambda_{rcon}$ are presented in Appendix D.5 and D.6, respectively.

### 4.4 Q3. Further Ablation Study

In this section, we carry out further ablation studies to see the effect of episodic incentive $r^p$ presented in Section 3.2. From EMU (QPLEX) and EMU (CDS), we ablate the episodic incentive and denote them with **(No-EI)**. We additionally ablate embedding network $f_\phi$ from EMU and denote them with **(No-SE)**. In addition, we ablate both parts, yielding EMC (QPLEX-original) and CDS (QPLEX-original). We evaluate the performance of each model on super hard SMAC maps. Additional ablation studies on GRF maps are presented in Appendix D.7. Note that EMC (QPLEX-original) utilizes the conventional episodic control presented in Zheng et al. (2021).

Figure 9 illustrates that the episodic incentive largely affects learning performance. Especially, EMU (QPLEX-No-EI) and EMU (CDS-No-EI) utilizing the conventional episodic control show a large performance variation according to different seeds. This demonstrates that a naive utilization of episodic control could be detrimental to learning an optimal policy. On the other hand, the episodic incentive selectively encourages transition considering desirability and thus prevents such a local convergence. Appendix D.9 and D.10 present an additional ablation study on semantic embedding

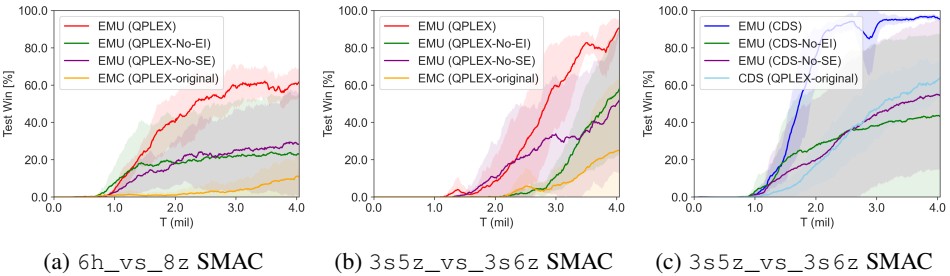

(a) 6h_vs_8z SMAC    (b) 3s5z_vs_3s6z SMAC    (c) 3s5z_vs_3s6z SMAC

Figure 9: Ablation studies on episodic incentive via complex MARL tasks.

and $r^c$, respectively. In addition, Appendix D.11 presents a comparison with an alternative incentive (Henaff et al., 2022) presented in a single-agent setting.

### 4.5 QUALITATIVE ANALYSIS AND VISUALIZATION

In this section, we conduct analysis with visualization to check how the desirability $\xi$ is memorized in $\mathcal{D}_E$ and whether it conveys correct information. Figure 10 illustrates two test scenarios with different seeds, and each snapshot is denoted with a corresponding timestep. In Figure 11, the trajectory of each episode is projected onto the embedded space of $\mathcal{D}_E$.

In Figure 10, case (a) successfully defeated all enemies, whereas case (b) lost the engagement. Both cases went through a similar, desirable trajectory at the beginning. For example, until $t = 10$ agents in both cases focused on killing one enemy and kept all ally agents alive at the same time. However, at $t = 12$, case (b) lost one agent, and two trajectories of case (a) and (b) in embedded space began to bifurcate. Case (b) still had a chance to win around $t = 14 \sim 16$. However,

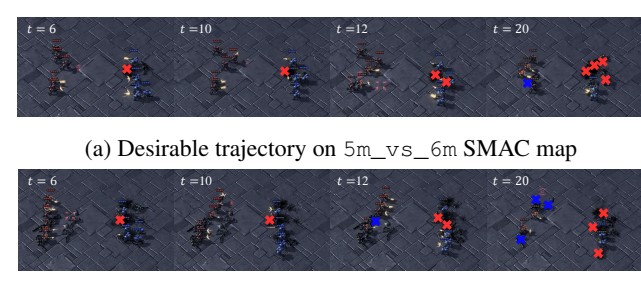

(a) Desirable trajectory on 5m_vs_6m SMAC map

(b) Undesirable trajectory on 5m_vs_6m SMAC map

Figure 10: Visualization of test episodes.

the states became *undesirable* (denoted without star marker) after losing three ally agents around $t = 20$, and case (b) lost the battle as a result. These sequences and characteristics of trajectories are well captured by desirability $\xi$ in $\mathcal{D}_E$ as illustrated in Figure 11.

Furthermore, the desirable state denoted with $\xi = 1$ encourages exploration around it though it is not directly retrieved during batch sampling. This occurs through the propagation of its desirability to states currently distinguished as undesirable during memory construction, using Algorithm 2 in Appendix E.1. Consequently, when the state's desirability is precisely memorized in $\mathcal{D}_E$, it can encourage desirable transitions through the episodic incentive $r^p$.

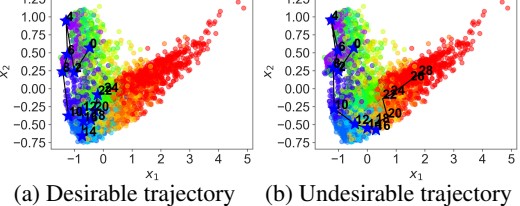

(a) Desirable trajectory    (b) Undesirable trajectory

Figure 11: Test trajectories on embedded space of $\mathcal{D}_E$.

## 5 CONCLUSION

This paper presents EMU, a new framework to efficiently utilize episodic memory for cooperative MARL. EMU introduces two major components: 1) a trainable semantic embedding and 2) an episodic incentive utilizing desirability of state. Semantic memory embedding allows us to safely utilize similar memory in a wide area, expediting learning via exploratory memory recall. The proposed episodic incentive selectively encourages desirable transitions and reduces the risk of local convergence by leveraging the desirability of the state. As a result, there is no need for manual hyperparameter tuning according to the complexity of tasks, unlike conventional episodic control. Experiments and ablation studies validate the effectiveness of each component of EMU.

ACKNOWLEDGEMENTS

This research was supported by AI Technology Development for Commonsense Extraction, Reasoning, and Inference from Heterogeneous Data(IITP) funded by the Ministry of Science and ICT(2022-0-00077).

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

# A    RELATED WORKS

This section presents the related works regarding incentive generation for exploration, episodic control, and the characteristics of cooperative MARL.

## A.1    INCENTIVE FOR MULTI-AGENT EXPLORATION

Balancing between exploration and exploitation in policy learning is a paramount issue in reinforcement learning. To encourage exploration, modified count-based methods (Bellemare et al., 2016; Ostrovski et al., 2017; Tang et al., 2017), prediction error-based methods (Stadie et al., 2015; Pathak et al., 2017; Burda et al., 2018; Kim et al., 2018), and information gain-based methods (Mohamed & Jimenez Rezende, 2015; Houthooft et al., 2016) have been proposed for a single agent reinforcement learning. In most cases, an incentive for exploration is introduced as an additional reward to a TD target in Q-learning; or such an incentive is added as a regularizer for overall loss functions. Recently, various aforementioned methods to encourage exploration have been adopted to the multi-agent setting (Mahajan et al., 2019; Wang et al., 2019; Jaques et al., 2019; Mguni et al., 2021) and have shown their effectiveness. MAVEN (Mahajan et al., 2019) introduces a regularizer maximizing the mutual information between trajectories and latent variables to learn a diverse set of behaviors. LIIR (Du et al., 2019) learns a parameterized individual intrinsic reward function by maximizing a centralized critic. CDS (Chenghao et al., 2021) proposes a novel information-theoretical objective to maximize the mutual information between agents' identities and trajectories to encourage diverse individualized behaviors. EMC (Zheng et al., 2021) proposes a curiosity-driven exploration by predicting individual Q-values. This individual-based Q-value prediction can capture the influence among agents as well as the novelty of states.

## A.2    EPISODIC CONTROL

Episodic control (Lengyel & Dayan, 2007) was well adopted on model-free setting (Blundell et al., 2016) by storing the highest return of a given state, to efficiently estimate its values or Q-values. Given that the sample generation is often limited by simulation executions or real-world observations, its sample efficiency helps to find an accurate estimation of Q-value (Blundell et al., 2016; Pritzel et al., 2017; Lin et al., 2018). NEC (Pritzel et al., 2017) uses a differentiable neural dictionary as an episodic memory to estimate the action value by the weighted sum of the values in the memory. EMDQN (Lin et al., 2018) utilizes a fixed random matrix to generate a state representation, which is used as a key to link between the state representation and the highest return of the state in the episodic memory. ERLAM (Zhu et al., 2020) learns associative memories by building a graphical representation of states in memory, and GEM (Hu et al., 2021) develops state-action values of episodic memory in a generalizable manner. Recently, EMC (Zheng et al., 2021) extended the approach of EMDQN to a deep MARL with curiosity-driven exploration incentives. EMC utilizes episodic memory to regularize policy learning and shows performance improvement in cooperative MARL tasks. However, EMC requires a hyperparameter tuning to determine the level of importance of the one-step TD memory-based target during training, according to the difficulties of tasks. In this paper, we interpret this regularization as an additional transition reward. Then, we present a novel form of reward, called episodic incentive, to selectively encourage the transition toward desired states, i.e., states toward a common goal in cooperative multi-agent tasks.

## A.3    COOPERATIVE MULTI-AGENT REINFORCEMENT LEARNING (MARL) TASK

In general, there is a common goal in cooperative MARL tasks, which guarantees the maximum return that can be obtained from the environment. Thus, there could be many local optima with high returns but not the maximum, which means the agents failed to achieve the common goal in the end. In other words, there is a distinct difference between the objective of cooperative MARL tasks and that of a single-agent task, which aims to maximize the return as much as possible without any boundary determining success or failure. Our desirability definition presented in Definition 1 in MARL setting becomes well justified from this view. Under this characteristic of MARL tasks, learning optimal policy often takes a long time and even fails, yielding a local convergence. EMU was designed to alleviate these issues in MARL.

## B   MATHEMATICAL PROOF

In this section, we present the omitted proofs of Theorem 1 and Theorem 2 as follows.

### B.1   PROOF OF THEOREM 1

*Proof.* The loss function of a conventional episodic control, $L_\theta^{EC}$, can be expressed as the weighted sum of one-step inference TD error $\Delta\varepsilon_{TD} = r(s,a) + \gamma\max_{a'}Q_{\theta-}(s',a') - Q_\theta(s,a)$ and MC inference error $\Delta\varepsilon_{EC} = Q_{EC}(s,a) - Q_\theta(s,a)$.

$$L_\theta^{EC} = (r(s,\boldsymbol{a}) + \gamma\max_{\boldsymbol{a}'}Q_{\theta-}(s',\boldsymbol{a}') - Q_\theta(s,\boldsymbol{a}))^2 + \lambda(Q_{EC}(s,\boldsymbol{a}) - Q_\theta(s,\boldsymbol{a}))^2, \quad (11)$$

where $Q_{EC}(s,\boldsymbol{a}) = r(s,\boldsymbol{a}) + \gamma H(s')$ and $Q_{\theta-}$ is the target network parameterized by $\theta^-$. Then, the gradient of $L_\theta^{EC}$ can be derived as

$$\nabla_\theta L_\theta^{EC} = -2\nabla_\theta Q_\theta(s,\boldsymbol{a})[(r(s,\boldsymbol{a}) + \gamma\max_{a'}Q_{\theta-}(s',\boldsymbol{a}') - Q_\theta(s,\boldsymbol{a})) + \lambda(Q_{EC}(s,\boldsymbol{a}) - Q_\theta(s,\boldsymbol{a}))]$$

$$= -2\nabla_\theta Q_\theta(s,\boldsymbol{a})(\Delta\varepsilon_{TD} + \lambda\Delta\varepsilon_{EC}). \quad (12)$$

Now, we consider an additional reward $r^{EC}$ for the transition to a conventional Q-learning objective, the modified loss function $L_\theta$ can be expressed as

$$L_\theta = (r(s,\boldsymbol{a}) + r^{EC}(s,\boldsymbol{a},s') + \gamma\max_{\boldsymbol{a}'}Q_{\theta-}(s',\boldsymbol{a}') - Q_\theta(s,\boldsymbol{a}))^2. \quad (13)$$

Then, the gradient of $L_\theta$ presented in Eq. 13 is computed as

$$\nabla_\theta L_\theta = -2\nabla_\theta Q_\theta(s,\boldsymbol{a})(\Delta\varepsilon_{TD} + r^{EC}). \quad (14)$$

Comparing Eq. 12 and Eq. 14, if we set the additional transition reward as $r^{EC}(s,\boldsymbol{a},s') = \lambda(r(s,\boldsymbol{a}) + \gamma H(s') - Q_\theta(s,\boldsymbol{a}))$, then $\nabla_\theta L_\theta = \nabla_\theta L_\theta^{EC}$ holds. □

### B.2   PROOF OF THEOREM 2

*Proof.* From Eq. 7, the value of $\hat{\eta}(s')$ can be expressed as

$$\hat{\eta}(s') = \mathbb{E}_{\pi_\theta}[\eta(s')] \simeq \frac{N_\xi(s')}{N_{call}(s')}\big(H(f_\phi(s')) - \max_{a'}Q_{\theta-}(s',a')\big)]. \quad (15)$$

When the joint actions from the current time follow the optimal policy, $\boldsymbol{a} \sim \pi_\theta^*$, the cumulative reward from $s'$ converges to $V^*(s')$, i.e., $H(f_\phi(s')) \to V^*(s')$. Then, every recall of $\hat{x}' = \mathrm{NN}(f_\phi(s')) \in \mathcal{D}_E$ guarantees the desirable transition, i.e., $\xi(s') = 1$, where $\mathrm{NN}(\cdot)$ represents a function for selecting the nearest neighbor. As a result, as $N_{call}(s') \to \infty$, $\frac{N_\xi(s')}{N_{call}(s')} \to 1$, yielding $\hat{\eta}(s') \simeq \frac{N_\xi(s')}{N_{call}(s')}\big(H(f_\phi(s')) - \max_{a'}Q_{\theta-}(s',a')\big) \to V^*(s') - \max_{\boldsymbol{a}'}Q_{\theta-}(s',\boldsymbol{a}')$. Then, the gradient signal with the episodic incentive $\nabla_\theta L_\theta^p$ becomes

$$\begin{aligned}
\nabla_\theta L_\theta^p &= -2\nabla_\theta Q_\theta(s,\boldsymbol{a})[\Delta\varepsilon_{TD} + r^p] \\
&= -2\nabla_\theta Q_\theta(s,\boldsymbol{a})[\Delta\varepsilon_{TD} + \gamma\hat{\eta}(s')] \\
&\simeq -2\nabla_\theta Q_\theta(s,\boldsymbol{a})[\Delta\varepsilon_{TD} + \gamma(V^*(s') - \max_{\boldsymbol{a}'}Q_{\theta-}(s',\boldsymbol{a}'))] \\
&= -2\nabla_\theta Q_\theta(s,\boldsymbol{a})[r(s,\boldsymbol{a}) + \gamma\max_{\boldsymbol{a}'}Q_{\theta-}(s',\boldsymbol{a}') - Q_\theta(s,\boldsymbol{a}) + \gamma(V^*(s') - \max_{\boldsymbol{a}'}Q_{\theta-}(s',\boldsymbol{a}'))] \\
&= -2\nabla_\theta Q_\theta(s,\boldsymbol{a})[r(s,\boldsymbol{a}) + \gamma V^*(s') - Q_\theta(s,\boldsymbol{a})] \\
&= \nabla_\theta L_\theta^*,
\end{aligned} \quad (16)$$

which completes the proof. □

In addition, when $\max_{\boldsymbol{a}'}Q_{\theta-}(s',\boldsymbol{a}')$ accurately estimates $V^*(s')$, the original TD-target is preserved as the episodic incentive becomes zero, i.e., $r^p \to 0$. Then with the properly annealed intrinsic reward $r^c$, the learning objective presented in Eq. 10 degenerates to the original Bellman optimality equation (Sutton & Barto, 2018). On the other hand, even though the assumption of $H(s') \to V^*(s')$ yields $\Delta\varepsilon_{EC} \to \Delta\varepsilon_{TD}^*$, $\nabla_\theta L_\theta^{EC}$ has an additional bias $\Delta\varepsilon_{TD}$ due to weighted sum structure presented in Eq. 3. Thus, $\nabla_\theta L_\theta^{EC}$ can converge to $\nabla_\theta L_\theta^*$ only when $\max_{\boldsymbol{a}'}Q_{\theta-}(s',\boldsymbol{a}') \to V^*(s')$ and $\lambda \to 0$ at the same time.

## C  IMPLEMENTATION AND EXPERIMENT DETAILS

### C.1  DETAILS OF IMPLEMENTATION

**Encoder and Decoder Structure**
As illustrated in Figure 1(c), we have an encoder and decoder structure. For an encoder $f_\phi$, we evaluate two types of structure, **EmbNet** and **dCAE**. For **EmbNet** with the learning objective presented in Eq. 4, two fully connected layers with 64-dimensional hidden state are used with ReLU activation function between them, followed by layer normalization block at the head. On the other hand, for **dCAE** with the learning objective presented in Eq. 5, we utilize a deeper encoder structure which contains three fully connected layers with ReLU activation function. In addition, **dCAE** takes a timestep $t$ as an input as well as a global state $s_t$. We set episodic latent dimension $\dim(x) = 4$ as Zheng et al. (2021).

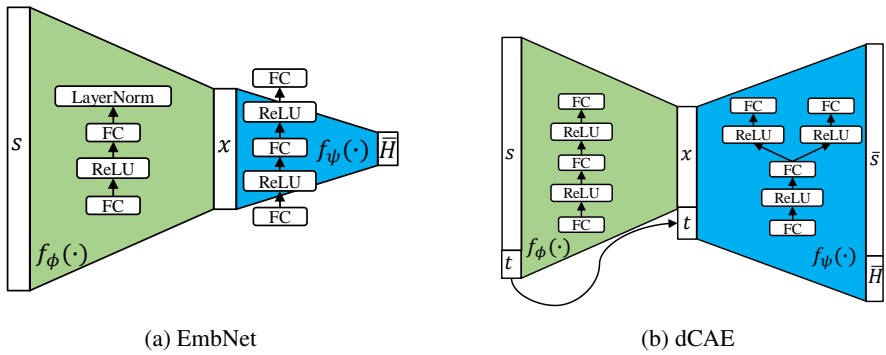

(a) EmbNet                              (b) dCAE

Figure 12: Illustration of network structures.

For a decoder $f_\psi$, both **EmbNet** and **dCAE** utilize a three-fully connected layer with ReLU activation functions. Differences are that **EmbNet** takes only $x_t$ as input and utilizes the 128-dimensional hidden state while **dCAE** takes $x_t$ and $t$ as inputs and adopts the 64-dimensional hidden state. As illustrated in Figure 1(c), to reconstruct global state $s_t$, **dCAE** has two separate heads while sharing lower parts of networks; $f_\phi^s$ to reconstruct $s_t$ and $f_\phi^H$ to predict the return of $s_t$, denoted as $H_t$. Figure 12 illustrates network structures of **EmbNet** and **dCAE**. The concept of supervised VAE similar to EMU can be found in (Le et al., 2018).

The reason behind avoiding probabilistic autoencoders such as variational autoencoder (VAE) (Kingma & Welling, 2013; Sohn et al., 2015) is that the stochastic embedding and the prior distribution could have an adverse impact on preserving a pair of $x_t$ and $H_t$ for given a $s_t$. In particular, with stochastic embedding, a fixed $s_t$ can generate diverse $x_t$. As a result, it breaks the pair of $x_t$ and $H_t$ for given $s_t$, which makes it difficult to select a valid memory from $\mathcal{D}_E$.

For training, we periodically update $f_\phi$ and $f_\psi$ with an update interval of $t_{emb}$ in parallel to MARL training. At each training phase, we use $M_{emb}$ samples out of the current capacity of $\mathcal{D}_E$, whose maximum capacity is 1 million (1M), and batch size of $m_{emb}$ is used for each training step. After updating $f_\phi$, every $x \in \mathcal{D}_E$ needs to be updated with updated $f_\phi$. Algorithm 1 shows the details of learning framework for $f_\phi$ and $f_\psi$. Details of the training procedure for $f_\phi$ and $f_\psi$ along with MARL training are presented in Appendix E.2.

**Other Network Structure and Hyperparameters**
For a mixer structure, we adopt QPLEX (Wang et al., 2020b) in both EMU (QPLEX) and EMU (CDS) and follow the same hyperparameter settings used in their source codes. Common hyperparameters related to individual Q-network and MARL training are adopted by the default settings of PyMARL (Samvelyan et al., 2019).

---

**Algorithm 1** Training Algorithm for State Embedding

---

1: **Parameter:** learning rate $\alpha$, number of training dataset $N$, batch size $B$
2: Sample Training dataset $(s^{(i)}, H^{(i)}, t^{(i)})_{i=1}^{N} \sim \mathcal{D}_E$,
3: Initialize weights $\phi, \psi \leftarrow \mathbf{0}$
4: **for** $i = 1$ to $\lfloor N/B \rfloor$ **do**
5:     Compute $(x^{(j)} = f_\phi(s^{(j)}|t^{(j)}))_{j=(i-1)B+1}^{iB}$
6:     Predict return $(\bar{H}^{(j)} = f_\psi^H(x^{(j)}|t^{(i)}))_{j=(i-1)B+1}^{iB}$
7:     Reconstruct state $(\bar{s}^{(j)} = f_\psi^s(x^{(j)})|t^{(i)})_{j=(i-1)B+1}^{iB}$
8:     Compute Loss $\mathcal{L}(\phi, \psi)$ via Eq. 5
9:     Update $\phi \leftarrow \phi - \alpha \frac{\partial \mathcal{L}}{\partial \phi}, \psi \leftarrow \psi - \alpha \frac{\partial \mathcal{L}}{\partial \psi}$
10: **end for**

---

## C.2 EXPERIMENT DETAILS

We utilize PyMARL (Samvelyan et al., 2019) to execute all of the baseline algorithms with their open-source codes, and the same hyperparameters are used for experiments if they are presented either in uploaded codes or in their manuscripts.

For a general performance evaluation, we test our methods on various maps, which require a different level of coordination according to the map's difficulties. Win-rate is computed with 160 samples: 32 episodes for each training random seed, and 5 different random seeds unless denoted otherwise.

Both the mean and the variance of the performance are presented for all the figures to show their overall performance according to different seeds. Especially for a fair comparison, we set $n_{\text{circle}}$, the number of training per a sampled batch of 32 episodes during training, as 1 for all baselines since some of the baselines increase $n_{\text{circle}} = 2$ as a default setting in their codes.

For performance comparison with baseline methods, we use their codes with fine-tuned algorithm configuration for hyperparameter settings according to their codes and original paper if available. For experiments on SMAC, we use the version of `starcraft.py` presented in RODE (Wang et al., 2021) adopting some modification for compatibility with QPLEX (Wang et al., 2020b). All SMAC experiments were conducted on StarCraft II version 4.10.0 in a Linux environment.

For Google research football task, we use the environmental code provided by (Kurach et al., 2020). In the experiments, we consider three official scenarios such as academy_3_vs_1_with_keeper (3_vs_1WK), academy_counterattack_easy (CA_easy), and academy_counterattack_hard (CA_hard).

In addition, for controlling $r^c$ in Eq. 10, the same hyperparameters related to curiosity-based (Zheng et al., 2021) or diversity-based exploration Chenghao et al. (2021) are adopted for EMU (QPLEX) and EMU (CDS) as well as for baselines EMC and CDS. After further experiments, we found that the curiosity-based $r^c$ from (Zheng et al., 2021) adversely influenced super hard SMAC task, with the exception of `corridor` scenario. Furthermore, the diversity-based exploration from Chenghao et al. (2021) led to a decrease in performance on both easy and hard SMAC maps. Thus, we decided to exclude the use of $r^c$ for EMU (QPLEX) on the super hard SMAC task and for EMU (CDS) on the easy/hard SMAC maps. EMU set task-dependent $\delta$ values as presented in Table 1. For other hyperparameters introduced by EMU, the same values presented in Table 8 are used throughout all tasks. For EMU (QPLEX) in `corridor`, $\delta = 1.3e - 5$ is used instead of $\delta = 1.3e - 3$.

Table 1: Task-dependent hyperparameter of EMU.

| Category | $\delta$ |
|---|---|
| easy/hard SMAC maps | $1.3e^{-5}$ |
| super hard SMAC maps | $1.3e^{-3}$ |
| GRF | $1.3e^{-3}$ |

## C.3 DETAILS OF MARL TASKS

In this section, we specify the dimension of the state space, the action space, the episodic length, and the reward of SMAC (Samvelyan et al., 2019) and GRF (Kurach et al., 2020).

In SMAC, the global state of each task of SMAC includes the information of the coordinates of all agents, and features of both allied and enemy units. The action space of each agent consists of the moving actions and attacking enemies, and thus it increases according to the number of enemies. The dimensions of the state and action space and the episodic length vary according to the tasks as shown in Table 2. For reward structure, we used the *shaped reward*, i.e., the default reward settings of SMAC, for all scenarios. The reward is given when dealing damage to the enemies and get bonuses for winning the scenario. The reward is scaled so that the maximum cumulative reward, $R_{max}$, that can be obtained from the episode, becomes around 20 (Samvelyan et al., 2019).

Table 2: Dimension of the state space and the action space and the episodic length of SMAC

| Task | Dimension of state space | Dimension of action space | Episodic length |
|------|--------------------------|---------------------------|-----------------|
| 1c3s5z | 270 | 15 | 180 |
| 3s5z | 216 | 14 | 150 |
| 3s_vs_5z | 68 | 11 | 250 |
| 5m_vs_6m | 98 | 12 | 70 |
| MMM2 | 322 | 18 | 180 |
| 6h_vs_8z | 140 | 14 | 150 |
| 3s5z_vs_3s6z | 230 | 15 | 170 |
| corridor | 282 | 30 | 400 |

In GRF, the state of each task includes information on coordinates, ball possession, and the direction of players, etc. The dimension of the state space differs among the tasks as in Table 3. The action of each agent consists of moving directions, different ways to kick the ball, sprinting, intercepting the ball and dribbling. The dimensions of the action spaces for each task are the same. Table 3 summarizes the dimension of the action space and the episodic length. In GRF, there can be two reward modes: one is "sparse reward" and the other is "dense reward." In sparse reward mode, agents get a positive reward +1 when scoring a goal and get -1 when conceding one to the opponents. In dense reward mode, agents can get positive rewards when they approach to opponent's goal, but the maximum cumulative reward is up to +1. In our experiments, we adopt sparse reward mode, and thus the maximum reward, $R_{max}$ is +1 for GRF.

Table 3: Dimension of the state space and the action space and the episodic length of GRF

| Task | Dimension of state space | Dimension of action space | Episodic length |
|------|--------------------------|---------------------------|-----------------|
| 3_vs_1WK | 26 | 19 | 150 |
| CA_easy | 30 | 19 | 150 |
| CA_hard | 34 | 19 | 150 |

## C.4 INFRASTRUCTURE

Experiments for SMAC (Samvelyan et al., 2019) are mainly carried out on NVIDIA GeForce RTX 3090 GPU, and training for the longest experiment such as `corridor` via EMU (CDS) took less than 18 hours. Note that when training is conducted with $n_{circle} = 2$, it takes more than one and a half times longer. Training encoder/decoder structure and updating $\mathcal{D}_E$ with updated $f_\phi$ together only took less than 2 seconds at most in `corridor` task. As we update $f_\phi$ and $f_\psi$ periodically with $t_{emb}$, the additional time required for a trainable embedder is certainly negligible compared to MARL training.

# D FURTHER EXPERIMENT RESULTS

## D.1 NEW PERFORMANCE INDEX

In this section, we present the details of a new performance index called *overall win-rate*, $\bar{\mu}_w$. For example, let $f_w^i(s)$ be the test win-rate at training time $s$ of $i$th seed run and $\mu_w^i(t)$ represents the time integration of $f_w^i(s)$ until $t$. Then, a normalized overall win-rate, $\bar{\mu}_w$, can be expressed as

$$\bar{\mu}_w(t) = \frac{1}{\mu_{\max}} \frac{1}{n} \sum_{i=1}^{n} \mu_w^i(t) = \frac{1}{\mu_{\max}} \frac{1}{n} \sum_{i=1}^{n} \int_0^t f_w^i(s)ds, \tag{17}$$

where $\mu_{\max} = t$ and $\bar{\mu}_w \in [0, 1]$.

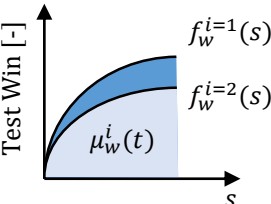

Figure 13: Illustration of $\mu_w^i(t)$.

Figure 13 illustrates the concept of time integration of win-rate, i.e., $\mu_w^i(t)$, to construct the overall win-rate, $\bar{\mu}_w$. By considering the integration of win-rate of each seed case, the performance variance can be considered, and thus $\bar{\mu}_w$ shows the training efficiency (speed) as well as the training quality (win-rate).

## D.2 ADDITIONAL EXPERIMENT RESULTS

In Section 4.3, we present the summary of parametric studies on $\delta$ with respect to various choices of $f_\phi$. To see the training efficiency and performance at the same time, Table 4 and 5 present the overall win-rate $\bar{\mu}_w$ according to training time. We conduct the experiments for 5 different seed cases and at each test phase 32 samples were used to evaluate the win-rate [%]. Note that we discard the component of episodic incentive $r^p$ to see the performance variations according to $\delta$ and types of $f_\phi$ more clearly.

Table 4: $\bar{\mu}_w$ according to $\delta$ and design choice of embedding function on **hard** SMAC map, `3s_vs_5z`.

| Training time [mil] | 0.69 | | | 1.37 | | | 2.00 | | |
|---|---|---|---|---|---|---|---|---|---|
| $\delta$ | random | EmbNet | dCAE | random | EmbNet | dCAE | random | EmbNet | dCAE |
| 1.3e-7 | 0.033 | 0.051 | **0.075** | 0.245 | 0.279 | **0.343** | 0.413 | 0.443 | **0.514** |
| 1.3e-5 | 0.010 | 0.044 | **0.063** | 0.171 | 0.270 | **0.325** | 0.320 | 0.441 | **0.491** |
| 1.3e-3 | 0.034 | 0.043 | **0.078** | 0.226 | 0.270 | **0.357** | 0.381 | 0.439 | **0.525** |
| 1.3e-2 | 0.019 | 0.005 | **0.079** | 0.205 | 0.059 | **0.346** | 0.348 | 0.101 | **0.518** |

Table 5: $\bar{\mu}_w$ according to $\delta$ and design choice of embedding function on **hard** SMAC map, `5m_vs_6m`.

| Training time [mil] | 0.69 | | | 1.37 | | | 2.00 | | |
|---|---|---|---|---|---|---|---|---|---|
| $\delta$ | random | EmbNet | dCAE | random | EmbNet | dCAE | random | EmbNet | dCAE |
| 1.3e-7 | 0.040 | **0.117** | 0.110 | 0.287 | 0.397 | **0.397** | 0.577 | 0.690 | **0.701** |
| 1.3e-5 | 0.064 | 0.107 | **0.131** | 0.334 | 0.402 | **0.436** | 0.634 | 0.714 | **0.749** |
| 1.3e-3 | 0.040 | **0.080** | 0.064 | 0.333 | **0.377** | 0.363 | 0.646 | **0.687** | 0.677 |
| 1.3e-2 | 0.038 | 0.000 | **0.048** | 0.288 | 0.001 | **0.332** | 0.584 | 0.001 | **0.643** |

As Table 4 and 5 illustrate that dCAE structure for $f_\phi$, which considers the reconstruction loss of global state $s$ as in Eq. 5, shows the best training efficiency in most cases. For `5m_vs_6m` task with $\delta = 1.3e^{-3}$, EmbNet achieves the highest value among $f_\phi$ choices in terms of $\bar{\mu}_w$ but fails to find optimal policy at $\delta = 1.3e^{-2}$ unlike other design choices. This implies that the reconstruction loss of dCAE facilitates the construction of a smoother embedding space for $\mathcal{D}_E$, enabling the retrieval of memories within a broader range of $\delta$ values from the key state. Figure 15 and 16 show the corresponding learning curves of each encoder structure for different $\delta$ values. A large $\delta$ value results in a higher performance variance than the cases with smaller $\delta$, according to different seed cases.

This is because a high value of $\delta$ encourages exploratory memory recall. In other words, by adjusting $\delta$, we can control the level of exploration since it controls whether to recall its own MC return or the highest value of other similar states within $\delta$. Thus, without constructing smoother embedding space as in dCAE, learning with exploratory memory recall within large $\delta$ can converge to sub-optimality as illustrated by the case of EmbNet in Figure 16(d).

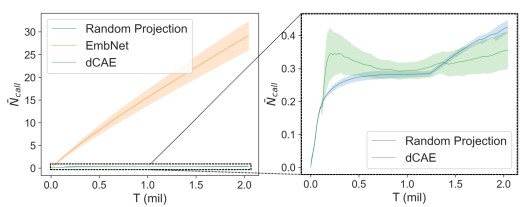

Figure 14: $\bar{N}_{call}$ of all memories in $\mathcal{D}_E$ when $\delta = 0.013$ according to design choice for $f_\phi$.

In Figure 14 which shows the averaged number of memory recall ($\bar{N}_{call}$) of all memories in $\mathcal{D}_E$, $\bar{N}_{call}$ of EmbNet significantly increases as training proceeds. On the other hand, dCAE was able to prevent this problem and recalled the proper memories in the early learning phase, achieving good training efficiency. Thus, embedding with dCAE can leverage a wide area of memory in $\mathcal{D}_E$ and becomes robust to hyperparameter $\delta$.

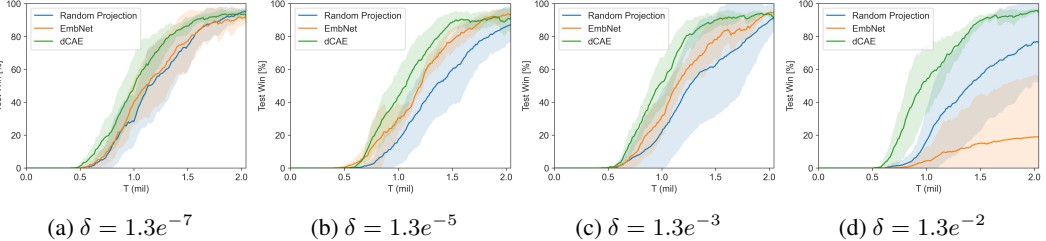

(a) $\delta = 1.3e^{-7}$     (b) $\delta = 1.3e^{-5}$     (c) $\delta = 1.3e^{-3}$     (d) $\delta = 1.3e^{-2}$

Figure 15: Parametric studies for $\delta$ on `3s_vs_5z` SMAC map.

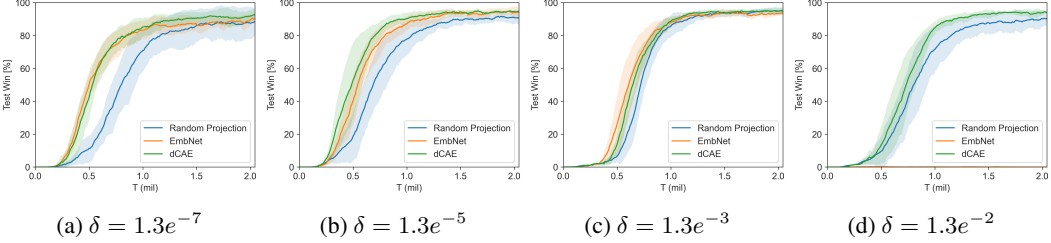

(a) $\delta = 1.3e^{-7}$     (b) $\delta = 1.3e^{-5}$     (c) $\delta = 1.3e^{-3}$     (d) $\delta = 1.3e^{-2}$

Figure 16: Parametric studies for $\delta$ on `5m_vs_6m` SMAC map.

### D.3 COMPARATIVE EVALUATION ON ADDITIONAL STARCRAFT II MAPS

Figure 17 presents a comparative evaluation of EMU with baseline algorithms on additional SMAC maps. Adopting EMU shows performance gain in various tasks.

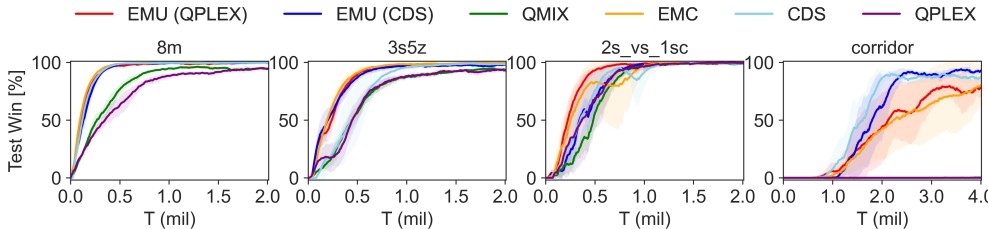

Figure 17: Performance comparison of EMU against baseline algorithms on additional SMAC maps.

### D.4 COMPARISON OF EMU WITH MAPPO ON SMAC

In this subsection, we compare the EMU with MAPPO (Yu et al., 2022) on selected SMAC maps. Figure 18 shows the performance in six SMAC maps: `1c3s5z`, `3s_vs_5z`, `5m_vs_6m`, `MMM2`, `6h_vs_8z` and `3s5z_vs_3s6z`. Similar to the previous performance evaluation in Figure 4, Win-rate is computed with 160 samples: 32 episodes for each training random seed and 5 different random seeds. Also, for MAPPO, scenario-dependent hyperparameters are adopted from their original settings in the uploaded source code.

From Figure 18, we can see that EMU performs better than MAPPO with an evident gap. Although after extensive training MAPPO showed a comparable performance against off-policy algorithm in its original paper (Yu et al., 2022), within the same training timestep used for our experiments, we found that MAPPO suffers from local convergence in super hard SMAC tasks such as `MMM2` and `3s5z_vs_3s6z` as shown in Figure 18. Only in `6h_vs_8z`, MAPPO shows comparable performance to EMU (QPLEX) with higher performance variance across different seeds.

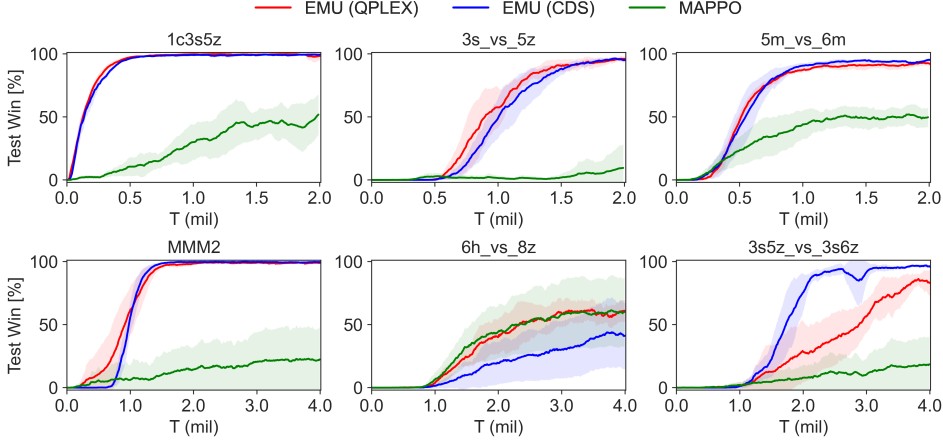

Figure 18: Performance comparison with MAPPO on selected SMAC maps.

### D.5 ADDITIONAL PARAMETRIC STUDY

In this subsection, we conduct an additional parametric study to see the effect of key hyperparameter $\delta$. Unlike the previous parametric study on Appendix D.2, we adopt both dCAE embedding network for $f_\phi$ and episodic reward. For evaluation, we consider three GRF tasks such as `academy_3_vs_1_with_keeper` (3_vs_1WK), `academy_counterattack_easy` (CA-easy), and `academy_counterattack_hard` (CA-hard); and one **super hard** SMAC map such as `6h_vs_8z`. For each task to evaluate EMU, four $\delta$ values, such as $\delta_1 = 1.3e^{-7}$, $\delta_2 = 1.3e^{-5}$, $\delta_3 = 1.3e^{-3}$, and $\delta_4 = 1.3e^{-2}$, are considred. Here, to compute the win-rate, 160 samples (32 episodes for each training random seed and 5 different random seeds) are used for 3_vs_1WK and 6h_vs_8z while 100 samples (20 episodes for each training random seed and 5 different random seeds) are used for CA-easy and CA-hard. Note that CDS and EMU (CDS) utilize the same hyperparameters, and EMC and EMU (QPLEX) use the same hyperparameters without a curiosity incentive presented in Zheng et al. (2021) as the model without it showed the better performance when utilizing episodic control.

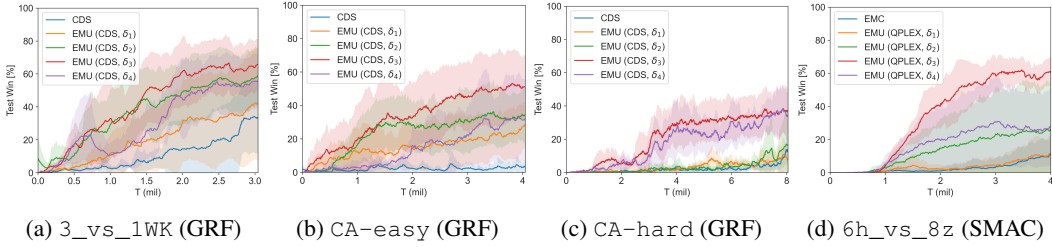

| (a) 3_vs_1WK (GRF) | (b) CA-easy (GRF) | (c) CA-hard (GRF) | (d) 6h_vs_8z (SMAC) |

Figure 19: Parametric studies for $\delta$ on various GRF maps and **super hard** SMAC map.

In all cases, EMU with $\delta_3 = 1.3e^{-3}$ shows the best performance. The tasks considered here are all complex multi-agent tasks, and thus adopting a proper value of $\delta$ benefits the overall performance and achieves the balance between exploration and exploitation by recalling the semantically similar memories from episodic memory. The optimal value of $\delta_3$ is consistent with the determination logic on $\delta$ in a memory efficient way presented in Appendix F.

### D.6 ADDITIONAL PARAMETRIC STUDY ON $\lambda_{rcon}$

Additionally, we conduct a parametric study for $\lambda_{rcon}$ in Eq. 5. For each task, EMU with five $\lambda_{rcon}$ values, such as $\lambda_{rcon,0} = 0.01$, $\lambda_{rcon,1} = 0.1$, $\lambda_{rcon,2} = 0.5$, $\lambda_{rcon,3} = 1.0$ and $\lambda_{rcon,4} = 10$, are evaluated. Here, to compute the win-rate of each case, 160 samples (32 episodes for each training random seed and 5 different random seeds) are used. From Figure 20, we can see that broad range of

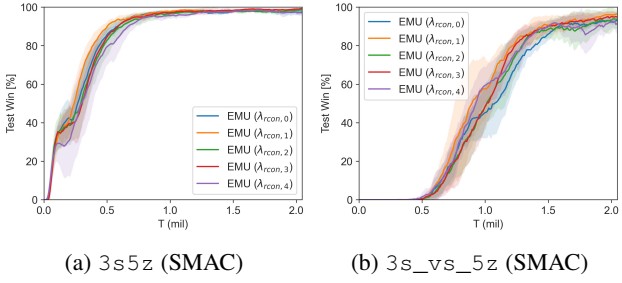

| (a) 3s5z (SMAC) | (b) 3s_vs_5z (SMAC) |

Figure 20: Parametric study for $\lambda_{rcon}$.

$\lambda_{rcon} \in \{0.1, 0.5, 1.0\}$ work well in general. However, with large $\lambda_{rcon}$ as $\lambda_{rcon,4} = 10$, we can observe that some performance degradation at the early learning phase in 3s5z task. This result is in line with the learning trends of Case 1 and Case 2 of 3s5z in Figure 23, which do not consider prediction loss and only take into account the reconstruction loss. Thus, considering both prediction loss and reconstruction loss as Case 4 in Eq. 5 with proper $\lambda_{rcon}$ is essential to optimize the overall learning performance.

## D.7 ADDITIONAL ABLATION STUDY IN GRF

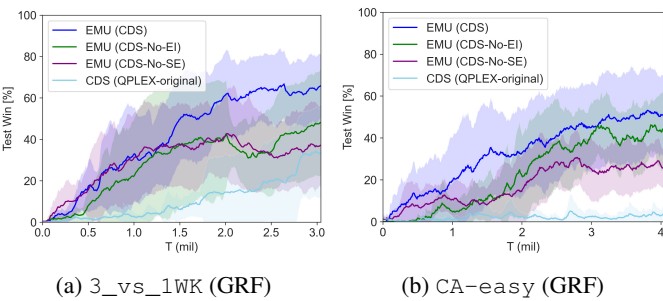

(a) `3_vs_1WK` (GRF)  (b) `CA-easy` (GRF)

Figure 21: Ablation studies on episodic incentive on GRF tasks.

In this subsection, we conduct additional ablation studies via GRF tasks to see the effect of episodic incentive. Again, EMU (CDS-No-EI) ablates episodic incentive from EMU (CDS) and utilizes the conventional episodic control presented in Eq. 3 instead. Again, EMU (CDS-No-SE) ablates semantic embedding by dCAE and adopts random projection with episodic incentive $r^p$. In both tasks, utilizing episodic memory with the proposed embedding function improves the overall performance compared to the original CDS algorithm. By adopting episodic incentives instead of conventional episodic control, EMU (CDS) achieves better learning efficiency and rapidly converges to optimal policies compared to EMU (CDS-No-EI).

## D.8 ADDITIONAL ABLATION STUDY ON EMBEDDING LOSS

In our case, the autoencoder uses the reconstruction loss to enforce the embedded representation $x$ to contain the full information of the original feature, $s$. We are adding $(H_t - f_\psi^H(f_\phi(s_t|t)|t))^2$ to guide the embedded representation to be consistent to $H_t$, as well, which works as a regularizer to the autoencoder. Therefore, $f_\psi^H$ is used in Eq. 5 to predict the observed $H_t$ from $D_E$ as a part of the semantic regularization effort.

Because $H_t$ is different from $f_\psi^H(x_t)$, the effort of minimizing their difference becomes the regularizer creating a gradient signal to learn $\psi$ and $\phi$. The update of $\phi$ results in the updated $x$ influenced by the regularization. Note that we update $\phi$ through the backpropagation of $\psi$.

The case of $L(\phi,\psi) = ||s_t - f_\psi^s(f_\phi(s_t|t)|t)||_2^2$ occurs when $\lambda_{rcon}$ becomes relatively much higher than 1, which makes $(H_t - f_\psi^H(f_\phi(s_t|t)|t))^2$ becomes ineffective. In other words, when $\lambda_{rcon}$ in Eq. 5 becomes relatively much higher than 1, $(H_t - f_\psi^H(f_\phi(s_t|t)|t))^2$ becomes ineffective.

The case of $L(\phi,\psi) = (H_t - f_\psi^H(f_\phi(s_t|t)|t))^2$ occurs when the scale factor $\lambda_{rcon}$ becomes relatively much smaller than 1, which makes $(H_t - f_\psi^H(f_\phi(s_t|t)|t))^2$ become a dominant factor. We conduct ablation studies considering four cases as follows:

- **Case 1:** $L(\phi,\psi) = ||s_t - f_\psi^s(f_\phi(s_t))||_2^2$, presented in Figure 22(a)
- **Case 2:** $L(\phi,\psi) = ||s_t - f_\psi^s(f_\phi(s_t|t)|t)||_2^2$, presented in Figure 22(b)
- **Case 3:** $L(\phi,\psi) = (H_t - f_\psi^H(f_\phi(s_t|t)|t))^2$, presented in Figure 22(c)
- **Case 4:** $L(\phi,\psi) = (H_t - f_\psi^H(f_\phi(s_t|t)|t))^2 + \lambda_{rcon}||s_t - f_\psi^s(f_\phi(s_t|t)|t)||_2^2$, i.e., Eq. 5, presented in Figure 22(d)

We visualize the result of t-SNE of 50K samples $x \in D_E$ out of 1M memory data trained by various loss functions: The task was 3s_vs_5z of SMAC as in Figure 2 and the training for all models proceeds for 1.5mil training steps. Case 1 and Case 2 showed irregular return distribution across the embedding space. In those two cases, there was no consistent pattern in the reward distribution. Case 3 with only return prediction in the loss showed better patterns compared to Case 1 and 2 but some features are not clustered well. We suspect that the consistent state representation also contributes to the return prediction. Case 4 of our suggested loss showed the most regular pattern in the return distribution arranging the low-return states as a cluster and the states with desirable returns as another

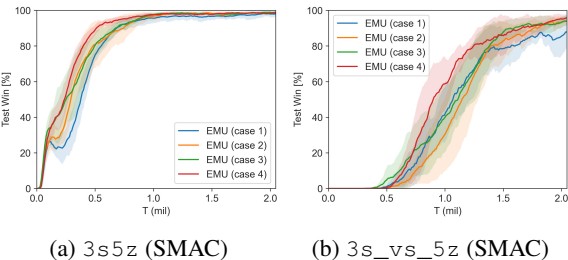

| (a) Loss (case 1) | (b) Loss (case 2) | (c) Loss (case 3) | (d) Loss (case 4) |

Figure 22: t-SNE of sampled embedding $x \in D_E$ trained by dCAE with various loss functions in `3s_vs_5z` SMAC map. Colors from red to purple represent from low return to high return.

cluster. In Figure 23, Case 4 shows the best performance in terms of both learning efficiency and terminal win-rate.

| (a) `3s5z` (SMAC) | (b) `3s_vs_5z` (SMAC) |

Figure 23: Performance comparison of various loss functions for dCAE.

## D.9 ADDITIONAL ABLATION STUDY ON SEMANTIC EMBEDDING

To further understand the role of semantic embedding, we conduct additional ablation studies and present them with the general performance of other baseline methods. Again, EMU (CDS-No-SE) ablates semantic embedding by dCAE and adopts random projection instead, along with episodic incentive $r^p$.

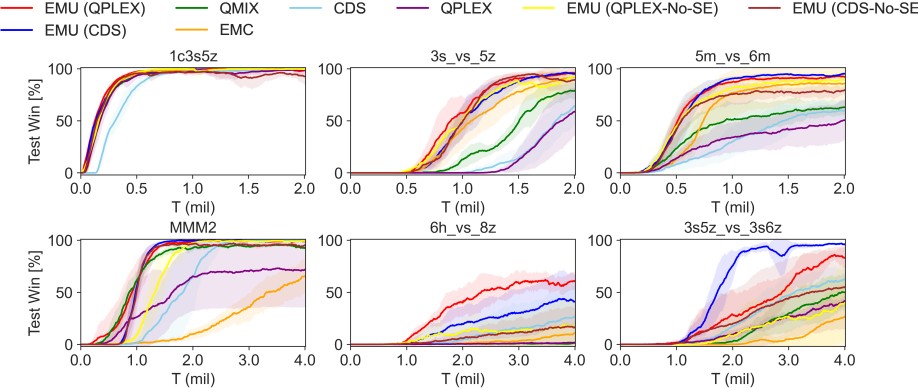

Figure 24: Performance comparison of EMU against baseline algorithms on three **easy and hard** SMAC maps: `1c3s5z`, `3s_vs_5z`, and `5m_vs_6m`, and three **super hard** SMAC maps: `MMM2`, `6h_vs_8z`, and `3s5z_vs_3s6z`.

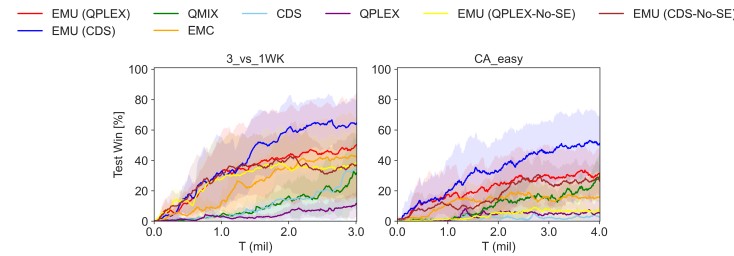

Figure 25: Performance comparison of EMU against baseline algorithms on Google Research Football.

For relatively easy tasks, EMU (QPLEX-No-SE) and EMU (CDS-No-SE) show comparable performance at first but they converge on sub-optimal policy in most tasks. Especially, this characteristic is well observed in the case of EMU (CDS-No-SE). As large size of memories are stored in an episodic buffer as training goes on, the probability of recalling similar memories increases. However, with random projection, semantically incoherent memories can be recalled and thus it can adversely affect the value estimation. We deem this is the reason for the convergence on suboptimal policy in the case of EMU (No-SE). Thus we can conclude that recalling semantically coherent memory is an essential component of EMU.

## D.10   ADDITIONAL ABLATION ON $r^c$

In Eq.10, we introduce $r^c$ as an additional reward which may encourage exploratory behavior or coordination. The reason we introduce $r^c$ is to show that EMU can be used in conjunction with any form of incentive encouraging further exploration. Our method may not be strongly effective until some desired states are found, although it has exploratory behavior via the proposed semantic embeddings, controlled by $\delta$. Until then, such incentives could be beneficial to find desired or goal states. Figures 26-27 show the ablation study of with and without $r^c$, and the contribution of $r^c$ is limited compared to $r^p$.

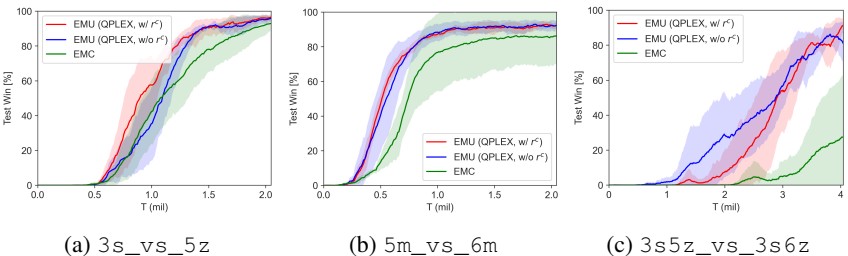

(a) 3s_vs_5z          (b) 5m_vs_6m          (c) 3s5z_vs_3s6z

Figure 26: Ablation studies on $r^c$ in SMAC tasks.

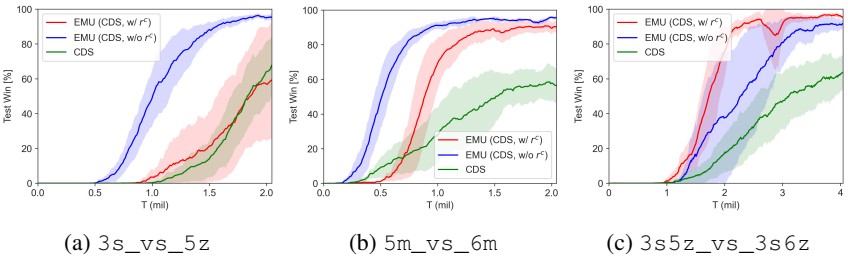

(a) 3s_vs_5z          (b) 5m_vs_6m          (c) 3s5z_vs_3s6z

Figure 27: Ablation studies on $r^c$ in SMAC tasks.

## D.11   COMPARISON OF EPISODIC INCENTIVE WITH EXPLORATORY INCENTIVE

In this subsection, we replace the episodic incentive with another exploratory incentive, introduced by (Henaff et al., 2022). In (Henaff et al., 2022), the authors extend the count-based episodic bonuses to continuous spaces by introducing episodic elliptical bonuses for exploration. In this concept, a high reward is given when the state projected in the embedding space is different from the previous states within the same episode. In detail, with a given feature encoder $\phi$, the elliptical bonus $b_t$ at timestep $t$ is computed as follows:

$$b_t = \phi(s_t)^T C_t^{-1} \phi(s_t) \tag{18}$$

where $C_t^{-1}$ is an inverse covariance matrix with an initial value of $C_{t=0}^{-1} = 1/\lambda_{e3b}\boldsymbol{I}$. Here, $\lambda_{e3b}$ is a covariance regularizer. For update inverse covariance, the authors suggested a computationally efficient update as

$$C_{t+1}^{-1} = C_t^{-1} - \frac{1}{1 + b_{t+1}} uu^T \tag{19}$$

where $u = C_t^{-1}\phi(s_{t+1})$. Then, the final reward $\bar{r}_t$ with episodic elliptical bonuses $b_t$ is expressed as

$$\bar{r}_t = r_t + \beta_{e3b}b_t \tag{20}$$

where $\beta_{e3b}$ and $r_t$ are a corresponding scale factor and external reward given by the environment, respectively.

For this comparison, we utilize the dCAE structure as a state embedding function $\phi$. For a mixer, QPLEX (Wang et al., 2020b) is adopted for all cases, and we denote the case with an elliptical incentive instead of the proposed episodic incentive as QPLEX (SE+E3B). Figure 28 illustrates

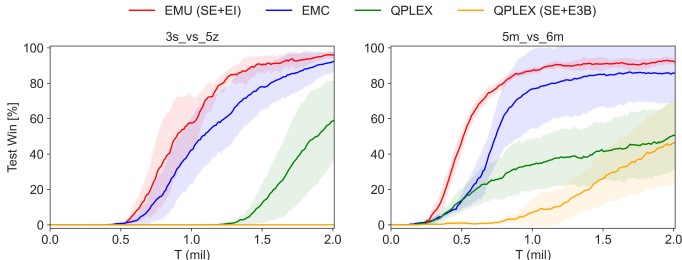

Figure 28: Performance comparison with elliptical incentive on selected SMAC maps.

the performance of adopting an elliptical incentive for exploration instead of the proposed episodic incentive. QPLEX (SE+E3B) uses the same hyperparameters with EMU (SE+EI) and we set $\lambda_{e3b} = 0.1$ according to Henaff et al. (2022).

As illustrated by Figure 28, adopting an elliptical incentive presented by (Henaff et al., 2022) instead of an episodic incentive does not give any performance gain and even adversely influences the performance compared to QPLEX. It seems that adding excessive surprise-based incentives can be a disturbance in MARL tasks since finding a new state itself does not guarantee better coordination among agents. In MARL, agents need to find the proper combination of joint action in a given similar observations when finding an optimal policy. On the other hand, in high-dimensional pixel-based single-agent tasks such as Habitat (Ramakrishnan et al., 2021), finding a new state itself can be beneficial in policy optimization. From this, we can note that adopting a certain algorithm from a single-agent RL case to MARL case may require a modification or adjustment with domain knowledge.

As a simple tuning, we conduct parametric study for $\beta_{e3b} = \{0.01, 0.1\}$ to adjust magnitude of incentive of E3B. Figure 29 illustrates the results. In Figure 29, QPLEX (SE+E3B) with $\beta_{e3b} = 0.01$ shows a better performance than the case with $\beta_{e3b} = 0.1$ and comparable performance to EMC in 5m_vs_6m. However, EMU with the proposed episodic incentive shows the best performance. From this comparison, we can see that incentives proposed by previous work need to be adjusted

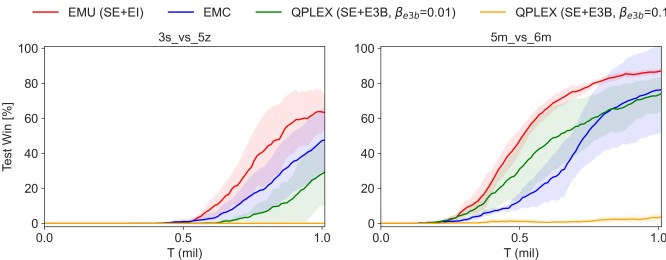

Figure 29: Performance comparison with an elliptical incentive on selected SMAC maps.

according to the type of tasks, as it was done in EMC (Zheng et al., 2021). On the other hand, with the proposed episodic incentive we do not need such hyperparameter-scaling, allowing much more flexible application across various tasks.

### D.12 Additional Toy Experiment and Applicability Tests

In this section, we conduct additional experiments on the didactic example presented by (Zheng et al., 2021) to see how the proposed method would behave in a simple but complex coordination task. Additionally, by defining $R_{thr}$ to define the desirability presented in Definition 1, we can extend EMU to a single-agent RL task, where a strict goal is not defined in general.

**Didactic experiment on Gridworld** We adopt the didactic example such as gridworld environment from (Zheng et al., 2021) to demonstrate the motivation and how the proposed method can overcome the existing limitations of the conventional episodic control. In this task, two agents in gridworld (see Figure 30(a)) need to reach their goal states at the same time to get a reward $r = 10$ and if only one arrives first, they get a penalty with the amount of $-p$. Please refer to (Zheng et al., 2021) for further details.

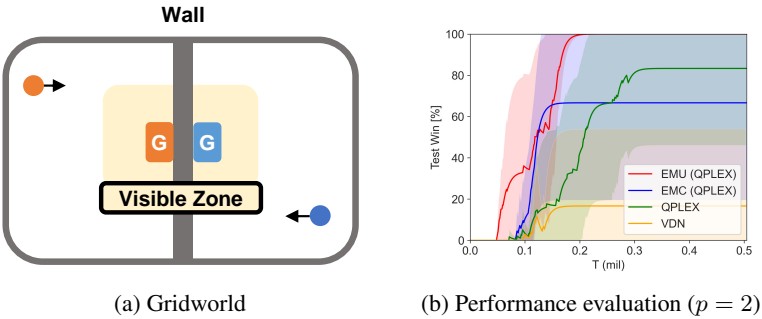

(a) Gridworld          (b) Performance evaluation ($p = 2$)

Figure 30: Didactic experiments on gridworld.

To see the sole effect of the episodic control, we discard the curiosity incentive part of EMC, and for a fair comparison, we set the same exploration rate of $\epsilon$-greedy with $T_\epsilon = 200K$ for all algorithms. We evaluate the win-rate with 180 samples (30 episodes for each training random seed and 6 different random seeds) at each training time. Notably, adopting episodic control with a naive utilization suffers from local convergence (see QPLEX and EMC (QPLEX) in Figure 30(b)), even though it expedites learning efficiency at the early training phase. On the other hand, EMU shows more robust performance under different seed cases and achieves the best performance by an efficient and discreet utilization of episodic memories.

**Applicability test to single agent RL task** We first need to define $R_{thr}$ value to effectively apply EMU to a single-agent task where a goal of an episode is generally not strictly defined, unlike cooperative multi-agent tasks with a shared common goal.

In a single-agent task where the action space is continuous such as MuJoCo (Todorov et al., 2012), the actor-critic method is often adopted. Efficient memory utilization of EMU can be used to train the critic network and thus indirectly influence policy learning, unlike general cooperative MARL tasks where value-based RL is often considered.

We implement EMU on top of TD3 and use the open-source code presented in (Fujimoto et al., 2018). We begin to train the model after sufficient data is stored in the replay buffer and conduct **6 times of training per episode with 256 mini-batches.** Note that this is different from the default settings of RL training, which conducts training at each timestep. Our modified setting aims to see the effect on the sample efficiency of the proposed model. The performance of the trained model is evaluated at every 50k timesteps.

We use the same hyperparameter settings as in MARL task presented in Table 8 except for the update interval, $t_{emb} = 100K$ according to large episodic timestep in single-RL compared to MARL tasks. It is worth mentioning that additional customized parameter settings for single-agent tasks may further improve the performance. In our evaluation, three single-agent tasks such as `Hopper-v4`, `Walker2D-v4` and `Humanoid-v4` are considered, and Figure 32 illustrates each task. Here, $\delta_2 = 1.3e - 5$ is used for `Hopper-v4` and `Walker2D-v4`, and $\delta_3 = 1.3e - 3$ is used for `Humanoid-v4` as `Humanoid-v4` task contains much higher state dimension space as 376-dimension. Please refer to Todorov et al. (2012) for a detailed description of tasks.

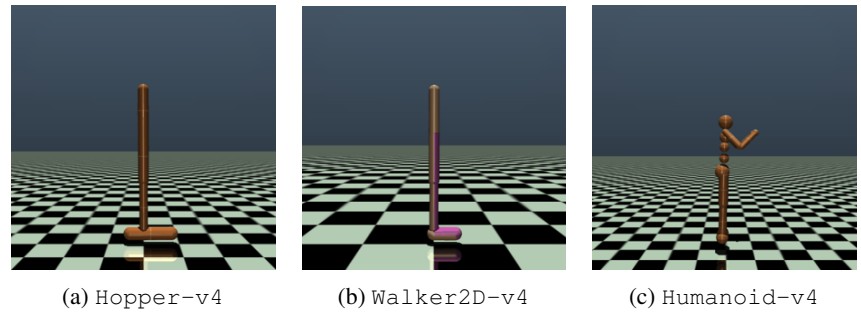

(a) `Hopper-v4`       (b) `Walker2D-v4`       (c) `Humanoid-v4`

Figure 31: Illustration of MuJoCo scenarios.

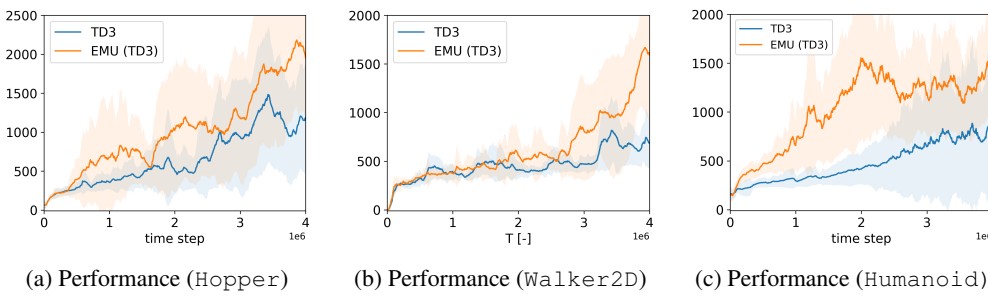

(a) Performance (`Hopper`)   (b) Performance (`Walker2D`)   (c) Performance (`Humanoid`)

Figure 32: Applicability test to single agent task ($R_{thr} = 500$).

In Figure 32, EMU (TD3) shows the performance improvement compared to the original TD3. Thanks to semantically similar memory recall and episodic incentive, states deemed desirable could have high values, and trained policy is encouraged to visit them more frequently. As a result, EMU (TD3) shows the better performance. Interestingly, under state dimension as `Humanoid-v4` task, TD3 and EMU (TD3) show a distinct performance gap in the early training phase. This is because, in a task with a high-dimensional state space, it is hard for a critic network to capture important features determining the value of a given state. Thus, it takes longer to estimate state value accurately. However, with the help of semantically similar memory recall and error compensation through episodic incentive, a critic network in EMU (TD3) can accurately estimate the value of the state much faster than the original TD3, leading to faster policy optimization.

Unlike cooperative MARL tasks, single-RL tasks normally do not have a desirability threshold. Thus, one may need to determine $R_{thr}$ based on domain knowledge or a preference for the level of return to be deemed successful. Figure 33 presents a performance variation according to $R_{thr}$.

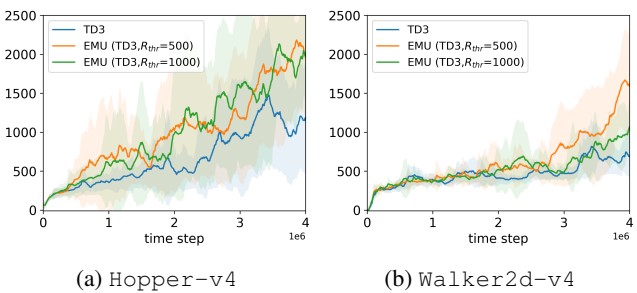

(a) `Hopper-v4`                (b) `Walker2d-v4`

Figure 33: Parametric study on $R_{thr}$.

When we set $R_{thr} = 1000$ in `Walker2d` task, desirability signal is rarely obtained compared to the case with $R_{thr} = 500$ in the early training phase. Thus, EMU with $R_{thr} = 500$ shows the better performance. However, both cases of EMU show better performance compared to the original TD3. In `Hopper` task, both cases of $R_{thr} = 500$ and $R_{thr} = 1000$ show the similar performance. Thus,

when determining $R_{thr}$, it can be beneficial to set a small value rather than a large one that can be hardly obtained.

Although setting a small $R_{thr}$ does not require much domain knowledge, a possible option to detour this is a periodic update of desirability based on the average return value $H(s)$ in all $s \in \mathcal{D}_E$. In this way, a certain state with low return which was originally deemed as *desirable* can be reevaluated as *undesirable* as training proceeds. The episodic incentive is not further given to those undesirable states.

**Scalability to image-based single-agent RL task** Although MARL tasks already contain high-dimension state space such as 322-dimension in `MMM2` and 282-dimension in `corridor`, image-based single RL tasks, such as Atari Bellemare et al. (2013) game, often accompany higher state spaces such as [210x160x3] for "RGB" and [210x160] for "grayscale". We use the "grayscale" type for the following experiments. For the details of the state space in MARL task, please see Appendix C.3.

In an image-based task, storing all state values to update all the key values in $\mathcal{D}_E$ as $f_\phi$ updates can be memory-inefficient, and a semantic embedding from original states may become overhead compared to the case without it. In such case, one may resort to a pre-trained feature extraction model such as ResNet model provided by torch-vision in a certain amount for dimension reduction only, before passing through the proposed semantic embedding. The feature extraction model above is not an object of training.

As an example, we implement EMU on the top of DQN model and compare it with the original DQN on Atari task. For the EMU (DQN), we adopt some part of pre-trained ResNet18 presented by torch-vision for dimensionality reduction, before passing an input image to semantic embedding. At each epoch, 320 random samples are used for training in `Breakout` task, and 640 random samples are used in `Alien` task. The same mini-batch size of 32 is used for both cases. For $f_\phi$ training, the same parameters presented in Table 8 are adopted except for the $t_{emb} = 10K$ considering the timestep of single RL task. We also use the same $\delta_2 = 1.3e - 5$ and set $R_{thr} = 50$ for `Breakout` and $R_{thr} = 40$ for `Alien`, respectively. Please refer to Bellemare et al. (2013) and `https://gymnasium.farama.org/environments/atari` for task details. As in Figure 34, we found a performance gain by adopting EMU on high-dimensional image-based tasks.

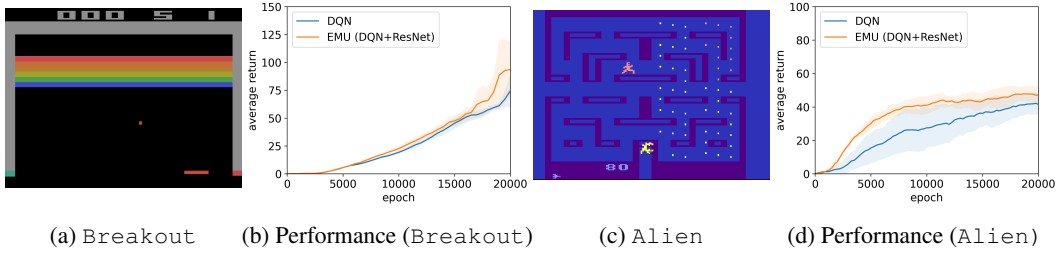

    (a) `Breakout`     (b) Performance (`Breakout`)     (c) `Alien`     (d) Performance (`Alien`)

Figure 34: Image-based single-RL task example.

# E   TRAINING ALGORITHM

## E.1   MEMORY CONSTRUCTION

During the centralized training, we can access the information on whether the episodic return reaches the highest return $R_{\max}$ or threshold $R_{thr}$, i.e., defeating all enemies in SMAC or scoring a goal in GRF. When storing information to $\mathcal{D}_E$, by the definition presented Definition. 1, we set $\xi(s) = 1$ for $\forall s \in \mathcal{T}_\xi$.

For efficient memory construction, we propagate the desirability of the state to a similar state within the threshold $\delta$. With this desirability propagation, similar states have an incentive for a visit. In addition, once a memory is saved in $\mathcal{D}_E$, the memory is preserved until it becomes obsolete (the oldest memory to be recalled). When a desirable state is found near the existing suboptimal memory within $\delta$, we replace the suboptimal memory with the desirable one, which gives the effect of a memory shift to the desirable state. Algorithm 2 presents the memory construction with the desirability propagation and memory shift.

---

**Algorithm 2** Episodic memory construction

---

1: $\xi_\mathcal{T}$: Optimality of trajectory
2: $\mathcal{T} = \{s_0, \boldsymbol{a_0}, r_0, s_1, ..., s_T\}$: Episodic trajectory
3: Initialize $R_t = 0$
4: **for** $t = T$ to $0$ **do**
5:     Compute $x_t = f_\phi(s_t)$ and $y_t = (x_t - \hat{\mu}_x)/\hat{\sigma}_x$
6:     pick the nearest neighbor $\hat{x}_t \in \mathcal{D}_E$ and get $\hat{y}_t$.
7:     **if** $||\hat{y}_t - y_t||_2 < \delta$ **then**
8:         $N_{call}(\hat{x}_t) \leftarrow N_{call}(\hat{x}_t) + 1$
9:         **if** $\xi_\mathcal{T} == 1$ **then**
10:             $N_\xi(\hat{x}_t) \leftarrow N_\xi(\hat{x}_t) + 1$
11:         **end if**
12:         **if** $\xi_t == 0$ and $\xi_\mathcal{T} == 1$ **then**            ▷ desirability propagation
13:             $\xi_t \leftarrow \xi_\mathcal{T}$                              ▷ memory shift
14:             $\hat{x}_t \leftarrow x_t, \hat{y}_t \leftarrow y_t, \hat{s}_t \leftarrow s_t$
15:             $\hat{H}_t \leftarrow R_t$
16:         **else**
17:             **if** $\hat{H}_t < R_t$ **then** $\hat{H}_t \leftarrow R_t$
18:             **end if**
19:         **end if**
20:     **else**
21:         Add memory $\mathcal{D}_E \leftarrow (x_t, y_t, R_t, s_t, \xi_t)$
22:     **end if**
23: **end for**

---

For memory capacity and latent dimension, we used the same values as Zheng et al. (2021), and Table 6 shows the summary of hyperparameter related to episodic memory.

Table 6: Configuration of Episodic Memory.

| Configuration | Value |
|---|---|
| episodic latent dimension, $\dim(x)$ | 4 |
| episodic memory capacity | 1M |
| a scale factor, $\lambda$ (for conventional episodic control only) | 0.1 |

The memory construction for EMU seems to require a significantly large memory space, especially for saving global states $s$. However, $\mathcal{D}_E$ uses CPU memory instead of GPU memory, and the memory required for the proposed embedder structure is minimal compared to the memory usage of original

Table 7: Additional CPU memory usage to save global states.

| SMAC task | CPU memory usage (1M data) (GiB) |
|---|---|
| 5m_vs_6m | 0.4 |
| 3s5z_vs_3s6z | 0.9 |
| MMM2 | 1.2 |

RL training (<1%). Thus, a memory burden due to a trainable embedding structure is negligible. Table 7 presents examples of CPU memory usage to save global states $s \in \mathcal{D}_E$.

### E.2 OVERALL TRAINING ALGORITHM

In this section, we present details of the overall MARL training algorithm including training of $f_\phi$. Additional hyperparameters related to Algorithm 1 to update encoder $f_\phi$ and decoder $f_\psi$ are presented in Table 8. Note that variables $N$ and $B$ are consistent with Algorithm 1.

Table 8: EMU Hyperparameters for $f_\phi$ and $f_\psi$ training.

| Configuration | Value |
|---|---|
| a scale factor of reconstruction loss, $\lambda_{rcon}$ | 0.1 |
| update interval, $t_{emb}$ | 1K |
| training samples, $N$ | 102.4K |
| batch size of training, $B$ | 1024 |

Algorithm 3 presents the pseudo-code of overall training for EMU. In Algorithm 3, network parameters related to a mixer and individual Q-network are denoted as $\theta$, and double Q-learning with target network is adopted as other baseline methods (Rashid et al., 2018; 2020; Wang et al., 2020b; Zheng et al., 2021; Chenghao et al., 2021).

---

**Algorithm 3** EMU: Efficient episodic Memory Utilization for MARL

1: $\mathcal{D}$: Replay buffer
2: $\mathcal{D}_E$: Episodic buffer
3: $Q_\theta^i$: Individual Q-network of $n$ agents
4: $M$: Batch size of RL training
5: Initialize network parameters $\theta, \phi, \psi$
6: **while** $t_{env} \leq t_{max}$ **do**
7:     Interact with the environment via $\epsilon$-greedy policy based on $[Q_\theta^i]_{i=1}^n$ and get a trajectory $\mathcal{T}$.
8:     Run Algorithm 2 to update $\mathcal{D}_E$ with $\mathcal{T}$
9:     Append $\mathcal{T}$ to $\mathcal{D}$
10:     **for** $k = 1$ **to** $n_{circle}$ **do**
11:         Get $M$ sample trajectories $[\mathcal{T}]_{i=1}^M \sim \mathcal{D}$
12:         Run MARL training algorithm using $[\mathcal{T}]_{i=1}^M$ and $\mathcal{D}_E$, to update $\theta$ with Eq.10
13:     **end for**
14:     **if** $t_{env}$ mod $t_{emb} == 0$ **then**
15:         Run Algorithm 1 to update $\phi, \psi$
16:         Update all $x \in \mathcal{D}_E$ with updated $f_\phi$
17:     **end if**
18: **end while**

---

Here, any CTDE training algorithm can be adopted for MARL training algorithm in `line 12` in Algorithm 3. As we mentioned in Section C.4, training of $f_\phi$ and $f_\psi$ and updating all $x \in \mathcal{D}_E$ only

takes less than two seconds at most under the task with largest state dimension such as `corridor`. Thus, the computation burden for trainable embedder is negligible compared to the original MARL training.

## F  MEMORY UTILIZATION

A remaining issue in utilizing episodic memory is how to determine a proper threshold value $\delta$ in Eq. 1. Note that this $\delta$ is used for both updating the memory and recalling the memory. One simple option is determining $\delta$ based on prior knowledge or experience, such as hyperparameter tuning. Instead, in this section, we present a more memory-efficient way for $\delta$ selection. When computing $||\hat{x} - x||_2 < \delta$, the similarity is compared elementwisely. However, this similarity measure puts a different weight on each dimension of $x$ since each dimension of $x$ could have a different range of distribution. Thus, instead of $x$, we utilize the normalized value. Let us define a normalized embedding $y$ with the statistical mean ($\mu_x$) and variance ($\sigma_x$) of $x$ as

$$y = (x - \mu_x)/\sigma_x. \tag{21}$$

Here, the normalization is conducted for each dimension of $x$. Then, the similarity measure via $||\hat{y} - y||_2 < \delta$ with Eq. 21 puts an equal weight to each dimension, as $y$ has a similar range of distribution in each dimension. In addition, an affine projection of Eq. 21 maintains the closeness of original $x$-distribution, and thus we can safely utilize $y$-distribution instead of $x$-distribution to measure the similarity.

In addition, $y$ defined in Eq. 21 nearly follows the normal distribution, although it does not strictly follow it. This is due to the fact that the memorized samples $x$ in $\mathcal{D}_E$ do not originate from the same distribution, nor are they uncorrelated, as they can stem from the same episode. However, we can achieve an approximate coverage of the majority of the distribution, specifically $3\sigma_y$ in both positive and negative directions of $y$, by setting $\delta$ as

$$\delta \leq \frac{(2 \times 3\sigma_y)^{\dim(y)}}{M}. \tag{22}$$

For example, when $M = 1e^6$ and $\dim(y) = 4$, if $\sigma_y \approx 1$ then $\delta \leq 0.0013$. This is the reason we select $\delta = 0.0013$ for the exploratory memory recall.

