# OpenReview forum: "Efficient Episodic Memory Utilization of Cooperative Multi-Agent Reinforcement Learning"
_ICLR.cc/2024/Conference — ICLR 2024 oral_

### Official Review · Reviewer_mSGj · 2023-10-27

**Soundness:** 3 good
**Presentation:** 3 good
**Contribution:** 2 fair
**Rating:** 6
**Confidence:** 3

**Summary:**

This paper aims to improve the efficiency in multi-agent reinforcement learning (MARL). It leverages episodic memory and introduces episodic incentive to help exploring desirable trajectory. This paper demonstrate both theoretical analyses and empirical results.

**Strengths:**

* This paper provides comprehensive theoretical analyses and strong performance improvement. The paper proves that the approach can help policies converge to the optimal policies. The paper also shows the great performance in Google football and StarCraft.
* This paper is well-structured and written. The paper provides full details about the method and the experiment. It also constructs detailed ablation studies.

**Weaknesses:**

* I have concerns about the desirable trajectory. In paper, the author set $R_{thr}=R_{max}$. Since the desirable trajectories are the states that can achieve maximum returns, they must be the optimal states. What if the agents are impossible to achieve $R_{max}$. How to determine $R_{thr}$ in other environments?
* The dimension of $x$ is very small (i.e. 4 according to table 4 in the appendix). It's doubtful that it can reconstruct the global state.

**Questions:**

* See weakness
* The approach adopts a state embedding instead of random projection. Does this make the approach more hard to converge?

---

> ### Author Response · Authors · 2023-11-18
> **Response to Reviewer mSGj**
>
> **Q1.** I have concerns about the desirable trajectory. In paper, the author set $R_{thr}=R_{max}$. Since the desirable trajectories are the states that can achieve maximum returns, they must be the optimal states. What if the agents are impossible to achieve $R_{max}$. How to determine $R_{thr}$ in other environments?
>
> **A1.** The reason we argue our contribution majorly lies on MARL community, especially cooperative MARL, is that in cooperative MARL settings, there is an explicit common goal that can determine the "desirability" of a trajectory. In other tasks where such an explicit goal is not presented, we need to determine $R_{thr}$ based on domain knowledge or a preference for the level of return to be deemed successful.
>
> Considering the case where desirability is not specifically determined, We additionally conduct a parametric study on a single-RL task to see the effect of $R_{thr}$ value. For further details, please refer to **Appendix D.9**.
>
> **Q2.** The dimension of $x$ is very small (i.e. 4 according to table 4 in the appendix). It's doubtful that it can reconstruct the global state.
>
> **A2.** A major objective of our embedding function is to construct semantically similar memories clustered close. This is quite different from the objective of a generative model whose main purpose is to reconstruct inputs. As presented in the manuscript, considering reconstruction loss shows its merits in semantic embedding with $\textrm{dim}(x)=4$.
>
> In addition, we use $\textrm{dim}(x)=4$ is for fair comparison with EMC, which uses $\textrm{dim}(x)=4$ in their episodic memory construction.
>
> **Q3.** The approach adopts a state embedding instead of random projection. Does this make the approach more hard to converge?
>
> **A3.** As the reviewer mentioned, semantic embedding requires additional training. However, adopting semantic embedding instead of random projection can improve the convergence of "policy" that we want to optimize because EMU guarantees the statistically coherent incentive toward desirable states.
>
> Also, we can check the results from Figures 15-16 and EMU (XXX-No-SE) in ablation studies presented in Figure 9 and Figure 22. For small $\delta$, adopting semantic embedding or random projection shows not much difference in terms of convergence, indirectly viewed from the performance variance. However, their convergence speed is distinctively different. In other words, semantic embedding shows a much faster convergence to optimal policy. For large $\delta$, the semantic embedding (dCAE) shows a better convergence performance than a random projection.

---

> > ### Comment · Reviewer_mSGj · 2023-11-21
> >
> > Thanks for the reply. I'll maintain my score.

---

> > > ### Author Response · Authors · 2023-11-22
> > > **Thank you**
> > >
> > > Thank you for your time and effort. Please let us know if you have further inquiries about the paper. Sincerely, Authors.

---

### Official Review · Reviewer_7Wwz · 2023-10-31

**Soundness:** 3 good
**Presentation:** 4 excellent
**Contribution:** 3 good
**Rating:** 6
**Confidence:** 5

**Summary:**

The paper introduces the Efficient episodic Memory Utilization (EMU) for cooperative multi-agent reinforcement learning (MARL). Addressing the challenges in MARL where agents often get trapped in local optima, EMU aims to accelerate learning by leveraging a semantically coherent episodic memory buffer and selectively promoting desirable transitions. EMU uses an encoder/decoder structure to train semantically coherent episodic memory and introduces an episodic incentive reward structure to enhance performance. The proposed method is evaluated on benchmarks like StarCraft II and Google Research Football, demonstrating its superiority over existing methods.

**Strengths:**

1. This paper is well motivated and well written. Particularly, its visual illustration of Figure 2 and 3 are helpful.
2. Although the proposed idea of semantically coherent memory looks simple to use an AE structure, it seems not being explored in multi-agent settings. One potential related work is generalized episodic memory, which can also be regarded semantically coherent episodic memory.
3. The episodic incentive is interesting and looks effective.
4. The proposed method shows strong empirical results. Its ablation studies are extensively conducted.
5. This paper conducts a sufficient review of related work and is well positioned.

**Weaknesses:**

1. EMU needs to set a return threshold to determine the desirability of a trajectory. This may require some domain knowledge to properly determine it, even when using R_{max}. This knowledge may partially explain its outperformance.
2. When the key encoder is updated, the proposed method needs to update all keys in the memory, which seems quite computationally intensive.
3. It may be interesting to compare the proposed method with MAPPO.

**Questions:**

1. Can the author explain how to set the return threshold for desirability in experiments? Is this threshold dynamic or fixed?
2. Is the incentive reward only effective when a trajectory is desirable? Does this mean episodic memory is useful only when a very good trajectory is explored, which can be hard?

---

> ### Author Response · Authors · 2023-11-18
> **Response to Reviewer 7Wwz**
>
> **Q1.** EMU needs to set a return threshold to determine the desirability of a trajectory. This may require some domain knowledge to properly determine it, even when using $R_{max}$. This knowledge may partially explain its outperformance.
>
> **A1.** First, thank you for the valuable comments. The reason we argue our major contribution lies on cooperative MARL is that in such tasks, the common goal is often explicitly specified such as scoring a goal in GRF or defeating all enemies in SMAC. By utilizing that signal, we can determine whether a trajectory is desirable or not and use that signal to prevent local convergence. However, if there is no such explicit-common goal, one may resort to a domain knowledge or a preference on the level of return to be deemed successful.
>
> **Q2.** When the key encoder is updated, the proposed method needs to update all keys in the memory, which seems quite computationally intensive.
>
> **A2.** We present the computational burden to update all key values in the episodic memory and to train $f_{\phi}$ and $f_{\phi}$ in **Appendix C.3**. Sorry for that if it was hard to find. In corridor task which has high-dimensional state space, the training and update only took less than 2 seconds at most, which is certainly negligible compared to MARL training.
>
> **Q3.** It may be interesting to compare the proposed method with MAPPO.
>
> **A3.** Since MAPPO relies on on-policy and other baseline algorithms including EMU are off-policy, it is difficult to set hyperparameters such as batch size, training iterations for a fair comparison. Thus, we work on the training framework and the corresponding hyperparameters, and if the time is available, we will present the result by the end of the rebuttal period.
>
> **(updated)** We present the performance comparison of EMU with MAPPO in **Appendix D.12**. Please check the revised manuscript.
>
> **Q4.** Can the author explain how to set the return threshold for desirability in experiments? Is this threshold dynamic or fixed?
>
> **A4.** In our experiment, it is fixed. In cooperative MARL task, "desirability" is often determined explicitly as there is a common goal such as scoring a goal or defeating all enemies. Thus, in the task of SMAC and GRF we just use "desirability" signal generated by the environment, i.e., "battle_won" signal from the environment to determine whether an episode has achieved a goal or not. This signal can be identically generated if one set $R_{thr}=R_{max}$. The reason we present the desirability as $R \geq R_{thr}$ is for general definition considering the case where the explicit goal is not presented as in single-agent case. For these cases, we need to define $R_{thr}$ value, where the domain knowledge or preference on the level of reward to achieve should be introduced. However, again, in general cooperative MARL setting where the common goal is explicitly stated, we do not have to determine $R_{thr}$ value. We additionally conduct a parametric study on a single-RL task to see the effect of $R_{thr}$ value, considering the case where desirability is not specifically determined. For further details, please refer to Appendix D.9.
>
> **Q5.** Is the incentive reward only effective when a trajectory is desirable? Does this mean episodic memory is useful only when a very good trajectory is explored, which can be hard?
>
> **A5.** The incentive is only given to states considered desirable. This mechanism prevents local convergence toward states that have a good return at the early training phase but are not optimal. As the reviewer mentioned, the episodic incentive works after finding desirable trajectories which can be done by standard exploration with $\epsilon$-greedy or some additional incentive such as $r^c$. However, the problem is that even after we find desirable trajectories, the previous work could not fully utilize those good memories. On the contrary, in EMU, thanks to the semantic embedding, we can utilize semantically similar memories and thus encourage some exploration toward states that currently have not achieved goals but have the potential to do so. In this way, EMU can encourage exploration toward desirable states, deemed undesirable without semantic embedding.

---

> > ### Comment · Reviewer_7Wwz · 2023-11-23
> >
> > I appreciate the detailed response. I will maintain my rating.

---

> > > ### Author Response · Authors · 2023-11-23
> > > **Thank you**
> > >
> > > Thank you for your time and effort. Please let us know if you have further inquiries about the paper. Sincerely, Authors.

---

### Official Review · Reviewer_t4Jc · 2023-10-31

**Soundness:** 3 good
**Presentation:** 3 good
**Contribution:** 3 good
**Rating:** 6
**Confidence:** 4

**Summary:**

This paper presents a new framework called Efficient episodic Memory Utilization (EMU) to effectively exploit episodic memory for cooperative multi-agent reinforcement learning (MARL). EMU mainly relies on two features:
 1) A learned semantic embedding embedding that allows to easily pair similar states
 2) An "episodic incentive" mechanism to select the most useful transitions from the buffer when learning

**Strengths:**

This paper presents a novel framework and studies the effect of episodic memory in MARL, which to the best of my knowledge is an underexplored area. The theoretical foundation is good although sometimes it is difficult to grasp the motivation and intuition of some parts when first presented. The paper also includes a sound analysis of the performance of EMU in two relevant MARL settings with abundant ablations that help to understand the contributions of the different features.

**Weaknesses:**

The biggest weakness of this work at its current state is the limited scope of the literature review and the lack of comparison with existing methods for episodic memory. There are multiple works ([1-3] to name a few) that have created similar frameworks in single-agent settings, specially in exploration settings. So one wonders what prevents port these frameworks here? Without that for instance the embedding procedure of EMU could be a reinventing the wheel from existing procedures in [1,2]. I strongly encourage authors to visit that line of works and contrast those approaches with the features incorporated in EMU.

Moreover, I believe that the comparison with related work is imperative to understand the position of the paper and its contributions and should not be relegated to the appendix.


As a minor issue, writing also should be reviewed, but clarity in general is good

[1] Henaff, M., Raileanu, R., Jiang, M., & Rocktäschel, T. (2022). Exploration via elliptical episodic bonuses. Advances in Neural Information Processing Systems, 35, 37631-37646.
[2] Le, H., Do, K., Nguyen, D., & Venkatesh, S. (2023). Intrinsic Motivation via Surprise Memory. arXiv preprint arXiv:2308.04836.
[3] Fernandes, D. M., Kaushik, P., Shukla, H., & Surampudi, B. R. (2022). Momentum Boosted Episodic Memory for Improving Learning in Long-Tailed RL Environments.



---- Post Second Rebuttal ---

I want to thank the authors for the additional work to address the concerns, specially what you wrote in your last response from
"It seems that adding a surprise-based incentive can be".... until "For these reasons, incorporating the feature embedding structure from the single-RL domain and its learning framework into the multi-agent domain may necessitate extensive modification, resulting in a distinct line of research." was the kind of motivation and comparison that I was looking for.

This, together with the results in Appendix D.13 makes clear that the existing methods from single agent literature is not as good as EMU in this context and that indeed it is a relevant contribution that authors are giving to the community.

My only remaining comment is that none of this new work and the discussion is present in the main document, (at least I don-t see anything in magenta there) I would encourage the authors to incorporate a reference in the introduction highlighting that existing methods in single agent reinforcement learning are not valid here (incorporating a reference to this part of the appendix),

I believe now that the paper is sound and there is enough evidence to support the main claims from the authors. I am updating my score accordingly.

**Questions:**

Beyond my recommendations above regarding writing, there is a common abuse of "the" through the text, e.g. "In spite of the required exploration in MARL with CTDE, ${the}$ recent works on episodic control emphasize the exploitation of episodic memory to expedite reinforcement learning. Episodic control (Lengyel & Dayan, 2007; Blundell et al., 2016; Lin et al., 2018; Pritzel et al., 2017) memorizes ${the}$ explored..."

---

> ### Author Response · Authors · 2023-11-18
> **Response to Reviewer t4Jc**
>
> **Q1. Regarding the contribution of EMU**
>
> **A1.** First, thank you for the comments. From a certain perspective, there are some commonalities between the listed references and EMU, such as episodic memory usage and embedding for representation learning, but their main objective of using episodic memory is a bit different.
>
> References presented in the manuscript aim to generate a better TD target by utilizing the values of state stored in episodic memory. The value in episodic memory can be deemed as the best return experienced throughout training. Aligned with previous works, our paper mainly aims to generate a correct TD target that could converge on a true TD target.
>
> On the other hand, the listed references by the reviewer majorly deal with exploration, as specifically mentioned by the reviewer. We contrast our paper with the listed references by the reviewer.
>
> [1] extends the count-based episodic bonuses to continuous spaces by introducing elliptical bonuses and encourages exploration with them. For feature embedding, [1] adopted an inverse dynamic model presented in [R1], to discard unnecessary state information for predicting the agent's action. [2] presents a noise-robust intrinsic reward, called surprise novelty, for exploration. [3] utilizes "familiarity" buffer to predict rare states and adopts contrastive momentum loss to prioritize long-tail states.
>
> However, we design the dCAE structure to extract the features that are critical in determining the value of a given state, i.e., semantic embedding. This semantic embedding allows us to recall similar memories in a wider range of episodic memory space, yielding "Efficient memory utilization." Implementation of the listed references does not give us such functionality. Moreover, EMU deals with the problems that might be caused by implementing the conventional episodic control in cooperative MARL settings, by introducing "desirability." With this desirability, EMU presents an episodic incentive that generates a correct TD target while preventing local convergence toward "undesirable states". None of the listed references [1-3] aim to correctly model the true TD target as we do.
>
> Considering the comments above, we hope the reviewer to see the differences between the listed references and EMU, and the contribution points raised by EMU.
>
> [R1] Pathak, Deepak, et al. "Curiosity-driven exploration by self-supervised prediction." International conference on machine learning. PMLR, 2017.
>
> **Q2.** Moreover, I believe that the comparison with related work is imperative to understand the position of the paper and its contributions and should not be relegated to the appendix.
>
> **A2.** We want to point out that adopting an algorithm from a single-agent to MARL is often non-trivial and thus the community recognizes it as a contribution point of EMC. Moreover, a naive adoptation can adversely influence the overall performance, as EMC did in the complex MARL tasks. From this view, this paper argues that our contribution majorly lies on MARL community where the desirability is well determined by whether the common goal is achieved or not, leaving the possibility of adopting EMU to single-agent tasks, as additional merit of EMU. Thus, we present that application for readers in the Appendix rather than the main manuscript. Further differences between single-RL and MARL tasks are presented in **Appendix A.3**.
>
> **Q3.** As a minor issue, writing also should be reviewed, but clarity in general is good. Beyond my recommendations above regarding writing, there is a common abuse of "the" through the text, e.g. "In spite of the required exploration in MARL with CTDE, the recent works on episodic control emphasize the exploitation of episodic memory to expedite reinforcement learning. Episodic control memorizes the explored..."
>
> **A3.** Thank you for your comment. We have revised the manuscript to address any awkward usages of the.

---

> > ### Comment · Reviewer_t4Jc · 2023-11-21
> > **Response to the authors**
> >
> > I want to thank to the authors for their response and the additional work in the appendix. However I am afraid that my concerns remain.
> >
> > Specifically, I agree with the authors that the works the works that I mentioned use memory embeddings towards exploration only and here that is just a subproblem, and yes, applying methods from single agent literature to multi-agent literature may be challenging on its own, I never said the opposite. However, just checking the first section of the paper, that is not what authors highlight as their main contributions, they just list "Episodic incentive generation" to do a better exploration of desired states and an "efficient memory embedding".
> >
> > The "Episodic incentive generation" is novel, although the analysis could have been improved by contrasting it with SOTA exploration methods as the ones I listed. However, if one of the main contributions according to the authors is to propose an "efficient memory embedding" the paper should compare their memory embedding mechanism with existing ones. Such comparison is still not present. Indeed, it seems that the method proposed here will be better suited for the problems studied here than the ones in previous works, but currently this work does not give a sound support for that claim.
> >
> > Additionally, as a more minor thing but yet relevant, I still believe that too much discussion about the position of this work and the conclusions is relegated to the appendix.

---

> > > ### Author Response · Authors · 2023-11-22
> > > **Additional response to Reviewer t4Jc**
> > >
> > > **Q1.** The "Episodic incentive generation" is novel, although the analysis could have been improved by contrasting it with SOTA exploration methods as the ones I listed.
> > >
> > > **A1.** We already compared EMU with SOTA MARL algorithm. Both EMC and CDS contain exploratory incentives.
> > >
> > > **Q2.** if one of the main contributions according to the authors is to propose an "efficient memory embedding" the paper should compare their memory embedding mechanism with existing ones. Such comparison is still not present.
> > >
> > > **A2.** If any of the listed references have already been well adopted for MARL settings or even for various single-RL tasks, we believe the proposition by the review is justifiable. However, this is not the case. Given that the review also concurs with adopting such algorithms in other fields of study, such as MARL, being non-trivial, we think omitting a comparison with such non-implemented baseline does not diminish the contribution of our works. Additionally, it is worth noting that the listed references primarily focus on the single-agent image-based domain, which involves extensive network manipulation for feature-based tasks in MARL.
> > >
> > > In addition, we also explain why the proposed dCAE structure and its loss function are effective throughout the paper and present comparisons with alternative embedding structures and loss functions in manuscript **Section 3.1**, **Appendix C.1, D.2, D.5, D.6.** Since the proposed dCAE structure regularized with the return value itself is also novel, we believe our work contributes to both state-embedding method and incentive generation.
> > >
> > > Even though we think such implementation and comparison are not critical in evaluating the contribution of EMU, we are working on adopting one of the exploration methods listed by the reviewer to MARL task. If time is available, we will present the experiment results by the end of the rebuttal period.
> > >
> > > **Q3.** Indeed, it seems that the method proposed here will be better suited for the problems studied here than the ones in previous works, but currently this work does not give a sound support for that claim.
> > >
> > > **A3.** Unlike the reviewer's comment, we coherently present the reasons why the proposed method works better:
> > >
> > > 1) Semantic embedding allows for semantically similar memory recall, enhancing sample efficiency beyond what previous works achieved. This is because states deemed undesirable can be reevaluated as desirable through memory construction, resulting in exploration toward them. Without semantic embedding, the performance degrades as presented in the ablation studies in **Section 4.4** and **Appendix D.7 and D.11**.
> > >
> > > 2) Episodic incentive selectively motivates the transition deemed "desirable", by considering the "desirability" of state, not solely relying on the stored return value. This approach helps prevent local convergence toward states with a high return that may not be optimal in the early training phase. Additionally, the proposed incentive generates an optimal gradient signal by **Theorem 2** and its magnitude is adaptively adjusted when the current Q-value is well estimated by target Q-network. This adaptive adjustment eliminates the need to fine-tune the additional incentive according to task level, as was done in previous work.
> > >
> > > **Q4.** As a more minor thing but yet relevant, I still believe that too much discussion about the position of this work and the conclusions is relegated to the appendix.
> > >
> > > **A4.** Due to the limited space of the manuscript, we have dedicated the main manuscript to the main contribution, the key concept of the proposed method and core experiments, leaving additional explanation and experiments to appendix for readers. We hope the review sees this as a valid excuse.

---

> ### Author Response · Authors · 2023-11-23
> **Additional response to Reviewer t4Jc (2)**
>
> First, thank you for the comment; it became clearer to see the benefits of EMU compared to methods possibly adoptable from a single-agent domain.
>
> To respond the reviewer's request, we conduct additional experiments to check whether an exploratory incentive proposed in single-agent domain, specifically E3B in [1], works in MARL domain. Please refer to **Appendix D.13** of the updated manuscript for experiment results. We found that replacing an episodic incentive of EMU with E3B-style exploratory incentive is not beneficial even after some hyperparameter tuning. With its original scale factor, the performance severely degrades.
>
> It seems that adding a surprise-based incentive can be a disturbance in MARL tasks since finding a new state itself does not guarantee a better coordination among agents. In MARL, agents need to find the proper combination of joint actions in similar observations when finding optimal policy. On the other hand, in high-dimensional pixel-based single-agent tasks such as Habitat [Ref1], finding a new state itself can be a beneficial in policy optimization. Without modification of algorithm, one may choose a different level of a scale factor, but a hyperparameter search according to different tasks can be a hurdle of adopting such algorithm to various tasks. On the other hand, the proposed episodic incentive does not require such hyperparameter scaling to adjust the magnitude of incentive as it automatically adjust the magnitude according to its convergence status. **This is one of merits of the proposed episodic incentive.**
>
> In addition, unfortunately, feature embedding structure presented in [1] is not directly applicable in MARL in following reasons:
>
> 1) There is joint action $a_{t}^{jt}$ in MARL while an inverse dynamics model $g$ used for training $\phi$ predicts a single action $a_t$ based on $\phi(s_t)$ and $\phi(s_{t+1})$ as $g(\phi(s_t),\phi(s_{t+1}))$.
>
> 2) In MARL, each agent's action $a_t^i$ is taken based on an agent-wise partial observation $o_t^i$. Thus, given global $s_t$ or $\phi(s_t)$ cannot properly predict $a_{t}^{jt}$. One could use partial observation as an input to $\phi$, as actionable representation learning adopted in [Ref2], but we want to learn $\phi(s_t)$ not $\phi(o_t^i)$.
>
> For these reasons, incorporating the feature embedding structure from the single-RL domain and its learning framework into the multi-agent domain may necessitate extensive modification, resulting in a distinct line of research.
>
> Considering the points mentioned above, we hope the reviewer understands the limitation of adopting algorithm from the listed references to MARL domain and sees the contribution points introduced by EMU.
>
> [1] Henaff, M., et al., "Exploration via elliptical episodic bonuses.", Advances in Neural Information Processing Systems, 35, 37631-37646, 2022.
>
> [Ref1] Ramakrishnan, Santhosh K., et al. "Habitat-matterport 3d dataset (hm3d): 1000 large-scale 3d environments for embodied ai." arXiv preprint arXiv:2109.08238 (2021).
>
> [Ref2] Jeon, Jeewon, et al. "Maser: Multi-agent reinforcement learning with subgoals generated from experience replay buffer." International Conference on Machine Learning. PMLR, 2022.

---

### Official Review · Reviewer_8Ngy · 2023-10-31

**Soundness:** 3 good
**Presentation:** 3 good
**Contribution:** 2 fair
**Rating:** 6
**Confidence:** 3

**Summary:**

This work presents a new framework for co-operative multi-agent RL that uses semantic memory embeddings to construct a novel reward structure that augments the environment reward by incentivizing desirable transitions. The framework, referred to as Efficient episodic Memory Utilization (EMU), comprises of an encoder-decoder network to learn semantically meaningful embeddings. The network is then utilized to obtain a reward that incentivizes desirable transitions. Experimental results in the benchmark Starcraft environments and Google Research Football demonstrate EMU’s superior performance to existing methods.

**Strengths:**

- **Clear writing and presentation:** Except for some minor subsections, the paper is generally well-written, easy to follow and presents a coherent story.
- **Promising and extensive results**: The method outperforms existing works in standard benchmark domains. The work presents many experiments and ablation studies to analyze and demonstrate the effectiveness of different components of the proposes framework.

**Weaknesses:**

- **Scalability**: Based on the encoder-decoder architecture of the paper, I am assuming that the global state for the environments used is feature-based. It is unclear whether this method will scale to vision-based environments due to **a)** the memory requirements of storing many images, **b)** the optimization difficulty in reconstructing image-based states and **c)** the effectiveness of the introduced reward structure in high-dimensional state spaces. However, I acknowledge that many existing works in this area utilize feature-based observations.
Regardless, it would greatly improve the strength of the results of this paper if some gains could be shown in vision-based environments.

- **Evaluation**: It appears that random projection performs almost equivalently to EmbNet/dCAE when compared using test win rates. The introduction of a new metric (overall win-rate) highlights that EmbNet/dCAE enable faster/more sample-efficient learning. However, improvement on this new metric is not as significant as the original win rate (which is the standard benchmark in the community). I would be curious to see the curve for EMU with random projection added to **Sections 4.1** and **4.2** to better understand the significance of EmbNet/dCAE.

**Questions:**

1. Results for **1)** in Weaknesses. These set of results are not completely necessary but would be good to see. A well-reasoned argument about why the method should not be difficult to scale will also suffice.

2. Results for **2)** in Weaknesses.

3. It would be helpful to provide details about the state space, action space, environment reward and episode lengths for both SMAC and GRF in the Appendix.

4. **Section 3.2** can use more intuition and better writing. It is unclear to me why the episodic inventive for a desirable transition is set to be proportional to the difference between the true value and the predicted value. Is this done to incentivize visits to states where the Q network has not converged?

5. What are the implications of *Theorem 2*?

---

> ### Author Response · Authors · 2023-11-18
> **Response to Reviewer 8Ngy**
>
> **Q1. Regarding scalability of EMU:** It is unclear whether this method will scale to vision-based environments due to a) the memory requirements of storing many images, b) the optimization difficulty in reconstructing image-based states and c) the effectiveness of the introduced reward structure in high-dimensional state spaces. However, I acknowledge that many existing works in this area utilize feature-based observations. Regardless, it would greatly improve the strength of the results of this paper if some gains could be shown in vision-based environments.
>
> **A1.** Thanks for the comment. First, we want to mention that our contribution lies majorly on applying episodic control method to cooperative MARL setting in a more efficient and robust way, by introducing a trainable semantic embedding and episodic incentive. As we evaluate EMU on multi-agent systems where high-dimensional state space, such as 322-dimension in MMM2 and 282-dimension in corridor, is already considered during centralized training, we deem our method also scalable to tasks with high-dimensional state spaces. In addition, episodic memory is saved in CPU rather than GPU, where the size RAM can be easily augmented.
>
> If still saving all states is impossible in a certain task or a semantic embedding from original states becomes overhead, one may resort to pre-trained feature extraction model such as ResNet model provided by torch-vision in a certain amount for dimension reduction only, before passing through the proposed semantic embedding.
>
> As an example, we implement EMU on the top of DQN model and compare it with original DQN on Atari task. For the EMU (DQN), we adopt some part of pre-trained ResNet18 presented by torch-vision for dimensionality reduction, before passing an input image to semantic embedding. We found a performance gain by adopting EMU on high-dimensional image-based tasks. Please refer to **Appendix D.9** for experimental details and the result.
>
> **Q2. Regarding the evaluation of EMU:** It appears that random projection performs almost equivalently to EmbNet/dCAE when compared using test win rates. The introduction of a new metric (overall win-rate) highlights that EmbNet/dCAE enable faster/more sample-efficient learning. However, improvement on this new metric is not as significant as the original win rate (which is the standard benchmark in the community). I would be curious to see the curve for EMU with random projection added to Sections 4.1 and 4.2 to better understand the significance of EmbNet/dCAE.
>
> **A2.** We partially agree with the reviewer's statement since a long convergence time is one of predicaments in MARL training. Thus, we want to introduce a metric that measures both learning speed and quality. Although their win-rates seem comparable in relatively easy tasks such as 3s\_vs\_5z and 5m\_vs\_6m, adopting random projection instead of semantic embedding degrades the performance as presented in Figure 9 and Figure 22 in the manuscript. Please see EMU (XXX-No-SE) case compared to EMU (XXX). We also conduct additional experiments on EMU (XXX-No-SE) in other tasks. For details, please refer to **Appendix D.11** of the revised manuscript.
>
> **Q3.** It would be helpful to provide details about the state space, action space, environment reward and episode lengths for both SMAC and GRF in the Appendix.
>
>  **A3.** Thanks for the comments, we present the details about the dimension of state space, action space, reward settings, and episode lengths for both SMAC and GRF in **Appendix D.10**. Please refer to it.

---

> > ### Author Response · Authors · 2023-11-18
> > **Response to Reviewer 8Ngy (2)**
> >
> > **Q4.** Section 3.2 can use more intuition and better writing. It is unclear to me why the episodic inventive for a desirable transition is set to be proportional to the difference between the true value and the predicted value. Is this done to incentivize visits to states where the Q network has not converged?
> >
> >  **A4.** More exactly, the proposed episodic incentive $r^p$ is given to states deemed desirable, i.e., feasible to achieve a common goal or $R_{thr}$. The incentive is not provided to undesirable states regardless of their Q-network convergence. Compared to the conventional episodic control, the proposed episodic incentive determines 1) which transition to give an incentive and 2) how much incentive to provide. Determining a desirable transition can be done by checking the desirability of the next state. A remaining issue is how much we incentivize those transitions. Arbitrary incentives make it hard to work well on various tasks since one may need to adjust the corresponding scale factor gradually during training or depending on the level of tasks.
> >
> > Instead, we developed an incentive whose magnitude is automatically adjusted during training, as $r^p$ proportional to the gap $H(f_{\phi}(s'))-max_{a'}Q_{\theta^-}(s',a')$. This gap compensates the underestimated value of $s'$ compared to $H(f_{\phi}(s'))$.
> >
> > Here, $H(f_{\phi}(s')) \rightarrow V^*(s')$ if $s'$ is desirable. However, a direct compensation can be too optimistic, thus we utilize the expected value of it by the count-based estimation with $N_{\xi}(s')/N_{call}(s')$. When $max_{a'}{Q_{\theta^-}}(s',a')$ accurately estimates $V^*(s')$, the original TD-target is preserved as the episodic incentive becomes zero, i.e., $r^p \rightarrow 0$. It is worth noting that the arbitrary magnitude of an incentive could hurt the convergence unless it converges to zero when $max_{a'}{Q_{\theta^-}}(s',a')$ already accurately estimates $V^*(s')$.
> >
> > Thus, the proposed episodic incentive gives an additional reward only to desired transition with the amount of discrepancy between true value and value estimated by target Q-network. In this way, we do not worry about the amount of incentive and a manual control for the corresponding scale factor as in the conventional episodic control. This is another key contribution of this paper in cooperative MARL settings.
> >
> > **Q5. What are the implications of Theorem 2?**
> >
> > **A5.** Through the proposed episodic incentive, we can generate a gradient signal as the optimal gradient for desired transitions. In other words, $r^p$ is designed to generate the correct gradient signal to the desirable transition, i.e., a transition toward $s'$ such that $\xi(s')=1$.

---

> > > ### Comment · Reviewer_8Ngy · 2023-11-21
> > > **Thank you**
> > >
> > > Thank you for taking the time to answer my questions, preparing additional results and modifying the paper to increase clarity. I appreciate the effort the authors put to address my concerns related to scalability and the semantic embeddings.
> > >
> > > Overall, I am happy with the work and have raised my score to reflect the same.

---

> > > > ### Author Response · Authors · 2023-11-22
> > > > **Thank you**
> > > >
> > > > Thank you for the valuable comments on the manuscript. Please let us know if you have further inquiries about the paper. Sincerely, Authors.

---

### Author Response · Authors · 2023-11-18
**Revised manuscript**

Dear reviewers,

First of all, we express our deepest gratitude for your constructive feedback and valuable comments. We add additional experiments requested by the reviewers in Appendix D.9, D.11 and D.12.

The revised parts of the manuscript are colored with **magenta**.

Sincerely, Authors.

---

> ### Author Response · Authors · 2023-11-21
> **Revised manuscript (2)**
>
> Dear Reviewers,
>
> Again, we would like to express our gratitude for the constructive comments you provided on our work. We genuinely appreciate the time and effort you have dedicated to this process. We uploaded the updated manuscript with our responses to address your comments. The revised or added parts of the manuscript are colored with **magenta**. If you require additional information or have any further inquiries about our paper, please let us know. We remain open to discussions and are ready to provide any necessary clarifications.
>
> Sincerely,
> Authors.

---

### Meta-Review · Area_Chair_TCpk · 2023-12-11

**Metareview:**

This paper introduces Efficient Episodic Memory Utilization (EMU) for cooperative multi-agent RL. EMU uses a semantic memory embeddings to create a unique reward structure that augments the traditional environment reward by promoting desirable transitions. The underlying model is an encoder-decoder network designed to learn semantically meaningful embeddings. These embeddings are then used to derive a reward that encourages desirable transitions.

EMU's effectiveness is demonstrated through extensive experiments and ablation studies in benchmark environments (Starcraft and Google Research Football). These results show that the proposed method outperforms baselines. There was also consensus that the paper is clearly written and everyone agreed that it is a meaningful contribution to an interesting problem.

One of the concerns was around scalability to other vision based environments but the authors responded back with reasonable explanations.

There was another issue raised by the reviewers regarding robustness of the eval metrics. The standard benchmark is typically original win rate but the authors proposed a new metric - overall win-rate and they use it to demonstrate that their method enables faster/more sample-efficient learning. The authors acknowledge the reviewer's observations on the similarity in win rates between random projection and EmbNet/dCAE in MARL training, and emphasize that their introduction of a new metric aims to evaluate both the speed and quality of learning. They point out that the superiority of semantic embedding over random projection is evident in more complex tasks (Figures 9 and 22, and further elaborated in Appendix D.11 of the revised manuscript).

Overall I think there is strong consensus that this method is novel, studies an interesting open problem in RL and conducted robust experimental validation on a range of tasks. Therefore I think the paper should be accepted.

**Justification For Why Not Higher Score:**

N/A

**Justification For Why Not Lower Score:**

Overall, there is a strong consensus that this method is novel, addresses an interesting open problem in RL, and has undergone robust experimental validation across a range of tasks. As RLHF techniques gain relevance in training multi-modal foundation models, and as teams of humans increasingly collaborate using these systems for real-world tasks, this line of research is likely to become quite important for future progress. Therefore, I believe the paper should be accepted for an oral presentation.

---

### Decision · Program_Chairs · 2024-01-16

Accept (oral)